CERN-TH-2023-090, PSI-PR-23-16, ZU-TH 23/23

# Two-Loop QCD Corrections for Three-Photon Production at Hadron Colliders

Samuel Abreu[1,2], Giuseppe De Laurentis[3], Harald Ita[3,4], Maximillian Klinkert[5], Ben Page[1] and Vasily Sotnikov[6,7]

**1** CERN, Theoretical Physics Department, CH-1211 Geneva 23, Switzerland
**2** Higgs Centre for Theoretical Physics, School of Physics and Astronomy,
The University of Edinburgh, Edinburgh EH9 3FD, Scotland, UK
**3** Paul Scherrer Institut, CH-5232 Villigen PSI, Switzerland
**4** ICS, University of Zurich, Winterthurerstrasse 190, 8057 Zurich, Switzerland
**5** Physikalisches Institut, Albert-Ludwigs-Universität Freiburg,
Hermann-Herder. Str. 3, D-79104 Freiburg, Germany
**6** Physik-Institut, University of Zurich,
Winterthurerstrasse 190, 8057 Zurich, Switzerland
**7** Department of Physics and Astronomy, Michigan State University,
567 Wilson Road, East Lansing, MI 48824, USA

October 16, 2023

## Abstract

We complete the computation of the two-loop helicity amplitudes for the production of three photons at hadron colliders, including all contributions beyond the leading-color approximation. We reconstruct the analytic form of the amplitudes from numerical finite-field samples obtained with the numerical unitarity method. This method requires as input surface terms for all relevant five-point non-planar integral topologies, which we obtain by solving the associated syzygy problem in embedding space. The numerical samples are used to constrain compact spinor-helicity ansätze, which are optimized by taking advantage of the known one-loop analytic structure. We make our analytic results available in a public C++ library, which is suitable for immediate phenomenological applications. We estimate that the inclusion of the subleading-color contributions will decrease the size of the two-loop corrections by about 30% to 50%, and the NNLO cross sections by a few percent, compared to the results in the leading-color approximation.

# 1 Introduction

With the rapid development of precision studies within the physics program at the Large Hadron Collider (LHC), there is a growing need for precise theory predictions for many Standard Model processes. The knowledge of higher-order radiative corrections in the strong coupling constant is essential for the correct interpretation of the associated experimental measurements, and provides better control over theoretical uncertainties. While many two-to-two reactions are now known at next-to-next-to-leading order (NNLO) in perturbation theory, NNLO theoretical predictions for two-to-three processes constitute the current state of the art.

The study of triphoton final states at hadron colliders such as the LHC offers excellent opportunities for several interesting investigations. For instance, it allows to explore the consequences of potentially anomalous gauge and Higgs couplings [1–5]. Furthermore, the study

of triphoton production is an important irreducible background process for the associated production of a photon with a beyond-the-Standard-Model particle that subsequently decays into a pair of photons [6–8]. Similar to diphoton production [9–11], triphoton production exhibits large next-to-leading order (NLO) and NNLO corrections [12–14]. Accordingly, NLO predictions significantly deviate from data [15], and NNLO is the first perturbative order providing reliable results [12, 13] (see also the related work of refs. [16, 17]). Thus, good control of NNLO QCD corrections to this process is of great importance.

While diphoton production has been known at NNLO QCD accuracy for more than a decade [9, 10], first results for triphoton production at the same level of accuracy have only recently been obtained [12,13]. This is due to the highly challenging nature of NNLO computations for five-particle scattering processes. It has been demonstrated that existing frameworks for the subtraction of infrared divergences at NNLO are, in principle, capable of handling arbitrary production process [13, 18–20]. However, the calculation of two-loop amplitudes for five-particle processes represents the current state of the art, and needs to be addressed on a case-by-case basis. While NNLO QCD corrections for most massless five-particle processes have already been considered [12, 13, 18, 19, 21–24], in most of these studies, including the NNLO QCD corrections to triphoton production [12,13], only the leading-color approximation of the double-virtual corrections has been employed. The first complete cross-section calculation that does not employ the leading-color approximation was performed in ref. [24], and a number of complete two-loop five-point amplitudes are also available [23–25].

Subleading-color corrections are more challenging to compute because they include non-planar Feynman graphs, which are notoriously more challenging to handle. In pure QCD, all non-planar contributions vanish in the limit of a large number of colors $N_c$ [26]. This still holds if the number of fermions flavors $N_f$ is considered to be of the order of $N_c$, i.e. when the ratio $N_f/N_c$ is kept constant as $N_c$ approaches infinity. The latter variant of the leading-color approximation is typically phenomenologically justified, provided errors of about 10% in the approximated contributions are deemed acceptable. On the other hand, terms originating from photons coupling to closed fermion loops involve nonplanar diagrams. While these terms are still suppressed in the *formal $N_c \to \infty$* limit, one might be concerned that their contribution is not suppressed numerically.

The goal of this work is to compute analytic expressions for all two-loop five-point amplitudes that contribute to the NNLO corrections for triphoton production at hadron colliders in massless QCD (ignoring top loops), and provide an efficient numerical implementation for use in phenomenological studies. Our calculation follows the multi-loop numerical unitarity approach [27–30] as implemented in CARAVEL [31]. In numerical unitarity, amplitudes are reduced to a set of master integrals by matching numerical evaluations of generalized unitarity cuts to a parametrization of the loop integrands. Analytic expressions can then be reconstructed using multivariate functional reconstruction techniques [32, 33] (see also refs. [34–38] for related developments). In order to perform our calculation within this framework, we make a number of theoretical developments.

Firstly, we develop a new approach to the problem of parametrising the integrand so that as many terms as possible vanish upon integration, due to integration-by-parts (IBP) identities [39, 40]. These terms are commonly referred to as *surface terms*, and it is a highly non-trivial problem to construct them in a form that enables efficient numerical evaluations. In this work, we present a novel method of deriving numerically efficient representations of surface terms. We achieve this by solving an associated syzygy problem [27, 41, 42] which, taking inspiration from refs. [43–45], is formulated in so-called embedding space. This allows us to identify a simpler *homogeneous* syzygy system, whose solutions we lift to full syzygies via linear algebra. By performing this calculation on a numerical phase-space point, we are able to construct "skeleton" syzygies, that allow us to numerically determine the full set of syzygies and

surface terms phase-space point by phase-space point in an efficient manner. This significantly reduces the expression size of surface terms, allowing us to efficiently match the integrand parametrisation to generalized unitarity cuts.

Secondly, we tackle the important problem of the large amount of samples required to perform analytic reconstruction. Indeed, while several improvements to the generic black-box reconstruction [32, 33] have been explored [23, 46, 47] over the years, cutting-edge calculations (see e.g. [47]) indicate that further developments will be important to tackle amplitudes with an increased number of scales. For this reason, we explore new techniques to construct ansätze whose analytic structure better exhibit the physical properties of the scattering amplitudes, performing the analytic reconstruction using spinor-helicity ansatz techniques [35, 36]. In contrast to a more traditional reconstruction using a set of independent kinematic invariants (as e.g. in [48]), this better manifests physical properties of the amplitudes. Combined with a further optimization of the ansatz based on the expectation that some features of the analytic structure of one-loop amplitudes are preserved at two loops, we find a significant reduction in the number of numerical samples that is required. In practice, we are able to reconstruct the most complicated helicity amplitude from only about 4000 evaluations, corresponding to an order of magnitude less than what was originally required for the reconstruction of the planar amplitudes in ref. [48].

Alongside this paper, we provide the analytic results for the complete two-loop triphoton production amplitudes in a collection of supplementary material. Furthermore, in order to facilitate the applicability of our results in phenomenological studies of triphoton production, we have implemented them in the efficient public C++ library `FivePointAmplitudes` [49]. This further allows us to analyze the important question of the impact of the subleading-color contributions on the two-loop corrections. Our study suggests that including these contributions will lead to a significant decrease in the size of the two-loop corrections, reducing them by approximately 30% to 50% compared to the results obtained in the leading-color approximation. This effect is larger than the corrections of about 10% expected from typical color-suppressed contributions. This confirms the concerns that nonplanar contributions arising from the photons coupling to closed fermion loops are not necessarily numerically suppressed. We note nevertheless that this substantial change in the two-loop corrections should be contrasted against the fact that the double-virtual contributions to the NNLO corrections to this process are observed to be small [12, 13].

The paper is organized as follows. In section 2 we classify the full set of gauge-invariant contributions to triphoton production. In section 3 we discuss our computational approach. We review numerical unitarity, discuss our approach to the construction of surface terms and how we construct compact spinor-helicity ansätze. In section 4, we discuss the structure of our results, their validation, and the format in which they are presented in ancillary files. We also showcase the numerical performance of our C++ implementation. In section 5, we discuss the impact of the subleading-color contributions on the double-virtual corrections. Finally, in section 6, we present our conclusions.

## 2  Notation and Conventions

We consider the $\mathcal{O}\!\left(\alpha_s^2\right)$ corrections to the production of three photons at hadron colliders. The loop-induced process $gg \to \gamma\gamma\gamma$ vanishes to all orders in the combined theory of QCD and QED due to charge-conjugation symmetry [50]. Therefore, the only contributing partonic process is

$$q(-p_1, -h_1) + \bar{q}(-p_2, -h_2) \to \gamma(p_3, h_3) + \gamma(p_4, h_4) + \gamma(p_5, h_5), \tag{1}$$

where $p_i$ and $h_i$ denote the momentum and the helicity of the $i^{\text{th}}$ particle, respectively. Throughout this paper, momenta and helicity labels are understood in the all-outgoing convention.

The process involves five massless particles. Thus, the underlying kinematic is specified by five Mandelstam invariants, which can be chosen to be

$$s_{12} = (p_1 + p_2)^2, \quad s_{23} = (p_2 + p_3)^2, \quad s_{34} = (p_3 + p_4)^2,$$
$$s_{45} = (p_4 + p_5)^2, \quad s_{15} = (p_1 + p_5)^2, \tag{2}$$

as well as the parity-odd contraction of four momenta,

$$\text{tr}_5 = \text{tr}(\gamma^5 \slashed{p}_1 \slashed{p}_2 \slashed{p}_3 \slashed{p}_4). \tag{3}$$

Strictly speaking, scattering amplitudes for processes such as that of eq. (1) cannot be expressed in terms of just the set $\{s_{12}, s_{23}, s_{34}, s_{45}, s_{51}, \text{tr}_5\}$, as this requires removing an arbitrary little-group-dependent factor. In fact, amplitudes depend not only on the four-momenta, but also on the helicities of the external states.

To better represent this dependence on the helicities, we can adopt a different set of variables, namely the two-component spinors, $\lambda_i^\alpha$ and $\tilde{\lambda}_i^{\dot{\alpha}}$, with $i \in \{1, \ldots, 5\}$. Starting from the $2 \times 2$ spinors $p_i^{\dot{\alpha}\alpha}$, which are given in terms of the respective four-momenta as $p_i^{\dot{\alpha}\alpha} = p_{i,\mu} \sigma^{\mu\dot{\alpha}\alpha}$, with $\sigma^{\mu\dot{\alpha}\alpha} = (\mathbb{1}, \vec{\sigma})$ and $\vec{\sigma}$ the Pauli matrices, the two-component spinors $\lambda_i^\alpha$ and $\tilde{\lambda}_i^{\dot{\alpha}}$ are defined by noting that for massless particles

$$\det(\{p_i^{\dot{\alpha}\alpha}\}) = 0 \quad \Longrightarrow \quad p_i^{\dot{\alpha}\alpha} = \tilde{\lambda}_i^{\dot{\alpha}} \lambda_i^\alpha. \tag{4}$$

Lowering of spinor indices is performed as $\lambda_{i,\alpha} = \epsilon_{\alpha\beta} \lambda_i^\beta$ and $\tilde{\lambda}_{i,\dot{\alpha}} = \epsilon_{\dot{\alpha}\dot{\beta}} \tilde{\lambda}_i^{\dot{\beta}}$, where we make use of the Levi-Civita symbol $\epsilon^{\alpha\beta} = \epsilon^{\dot{\alpha}\dot{\beta}} = -\epsilon_{\alpha\beta} = -\epsilon_{\dot{\alpha}\dot{\beta}} = i\sigma_2$. Invariant contractions of spinors give so-called spinor brackets, which we define as

$$\langle ij \rangle = \lambda_i^\alpha \lambda_{j,\alpha} \quad \text{and} \quad [ij] = \tilde{\lambda}_{i,\dot{\alpha}} \tilde{\lambda}_j^{\dot{\alpha}}. \tag{5}$$

These are related to the Mandelstam invariants in eq. (2) through $s_{ij} = \langle ij \rangle [ji]$. We also use longer spinor contractions, in particular

$$\langle i|j+k|i] = \langle ij \rangle [ji] + \langle ik \rangle [ki]. \tag{6}$$

Finally, we can express $\text{tr}_5$ as a polynomial in spinor brackets as

$$\text{tr}_5 = [12]\langle 23 \rangle [34]\langle 41 \rangle - \langle 12 \rangle [23]\langle 34 \rangle [41]. \tag{7}$$

**Helicity amplitudes**   We closely follow the notation and conventions of ref. [48], and denote the (renormalised) amplitudes for this process by

$$\mathcal{M}(1_q^{h_1}, 2_{\bar{q}}^{h_2}, 3_\gamma^{h_3}, 4_\gamma^{h_4}, 5_\gamma^{h_5}) := e_q^3 \delta_{i_1 i_2} \mathcal{A}(1_q^{h_1}, 2_{\bar{q}}^{h_2}, 3_\gamma^{h_3}, 4_\gamma^{h_4}, 5_\gamma^{h_5}), \tag{8}$$

where $i_1$ and $i_2$ are the color indices of the external quarks and $e_q$ is their electric charge. We call $\mathcal{A}(1_q^{h_1}, 2_{\bar{q}}^{h_2}, 3_\gamma^{h_3}, 4_\gamma^{h_4}, 5_\gamma^{h_5})$ the *helicity amplitudes* for the process in eq. (1), and will often suppress their arguments for simplicity.

Helicity amplitudes satisfy relations under permutations of the photon momenta or under charge and parity conjugation. For the process in eq. (1) there are two independent helicity configurations, which we choose to be

$$\mathcal{A}_{+++}(1, 2, 3, 4, 5) := \mathcal{A}(1_q^+, 2_{\bar{q}}^-, 3_\gamma^+, 4_\gamma^+, 5_\gamma^+),$$
$$\mathcal{A}_{-++}(1, 2, 3, 4, 5) := \mathcal{A}(1_q^+, 2_{\bar{q}}^-, 3_\gamma^-, 4_\gamma^+, 5_\gamma^+), \tag{9}$$

where we indexed the independent amplitudes by the photon helicities. We work in the 't Hooft–Veltman scheme of dimensional regularisation, setting the space-time dimensions to $D = 4 - 2\epsilon$, and use the definition of dimensionally regularised helicity amplitudes with external quarks given in ref. [46]. We perform the UV renormalisation in the $\overline{\text{MS}}$ scheme, where the amplitudes admit an expansion in terms of the renormalised QCD coupling constant $\alpha_s$ of the form

$$\mathcal{A} = \mathcal{A}^{(0)} + \frac{\alpha_s}{2\pi}\mathcal{A}^{(1)} + \left(\frac{\alpha_s}{2\pi}\right)^2 \mathcal{A}^{(2)} + \dots . \tag{10}$$

The coupling $\alpha_s$ is related to the bare coupling $\alpha_s^0$ through

$$\alpha_s^0 \mu_0^{2\epsilon} S_\epsilon = \alpha_s \mu^{2\epsilon}\left(1 - \frac{\beta_0}{\epsilon}\frac{\alpha_s}{2\pi} + \mathcal{O}\left(\alpha_s^2\right)\right), \quad S_\epsilon = (4\pi)^\epsilon e^{-\epsilon\gamma_E}, \tag{11}$$

where $\gamma_E$ is the Euler–Mascheroni constant, $\mu_0$ and $\mu$ are the dimensional regularization and renormalization scales (which we assume to be equal), and $\beta_0$ is the first coefficient of the QCD $\beta$-function,

$$\beta_0 = \frac{11}{6}C_A - \frac{2}{3}T_F N_f . \tag{12}$$

Here, $C_A = N_c$ is the quadratic Casimir of the adjoint representation of the $SU(N_c)$ group, $N_f$ is the number of massless quarks, and $T_F = 1/2$ is the normalization of the generators of the fundamental representation. Below we will also need the quadratic Casimir of the fundamental representation, $C_F = \frac{N_c^2 - 1}{2N_c}$. The coefficients of the perturbative expansion of the renormalised amplitudes $\mathcal{A}^{(\ell)}$ are related to their bare counterparts $\mathcal{A}_\mathcal{B}^{(\ell)}$, which are coefficients in a perturbative expansion in powers of $\alpha_s^0$, by

$$\mathcal{A}_\mathcal{B}^{(0)} = \mathcal{A}^{(0)}, \quad \mathcal{A}_\mathcal{B}^{(1)} = S_\epsilon \mathcal{A}^{(1)}, \quad \mathcal{A}_\mathcal{B}^{(2)} = S_\epsilon^2\left(\mathcal{A}^{(2)} + \frac{\beta_0}{\epsilon}\mathcal{A}^{(1)}\right). \tag{13}$$

The coefficients $\mathcal{A}^{(\ell)}$ can be decomposed into individually gauge-invariant contributions that scale differently with the number of light quarks $N_f$, the number of colors $N_c$ and the electric charges of the light fermions. Up to second order, we have [51, 52]

$$\mathcal{A}^{(1)} = C_F A^{(1)},$$

$$\mathcal{A}^{(2)} = C_F^2 B^{(2,0)} + C_F C_A B^{(2,1)} + C_F T_F N_f A^{(2,N_f)} + C_F T_F \left(\sum_{f=1}^{N_f} Q_f^2\right) A^{(2,\tilde{N}_f)}, \tag{14}$$

where $Q_f$ denotes the ratio of the charges of the light fermions running in a closed fermion loop to the charge of the initial state quark/anti-quark pair. We can rearrange eq. (14) as

$$\mathcal{A}^{(2)} = \frac{N_c^2}{4}\left(A^{(2,0)} - \frac{1}{N_c^2}(A^{(2,0)} + A^{(2,1)}) + \frac{1}{N_c^4}A^{(2,1)}\right)$$

$$+ C_F T_F N_f A^{(2,N_f)} + C_F T_F \left(\sum_{f=1}^{N_f} Q_f^2\right) A^{(2,\tilde{N}_f)}, \tag{15}$$

where we have

$$A^{(2,0)} := B^{(2,0)} + 2B^{(2,1)}, \qquad A^{(2,1)} := B^{(2,0)}. \tag{16}$$

The contributions $A^{(2,0)}$ and $A^{(2,N_f)}$ involve only planar diagrams and were previously computed in refs. [12, 48, 53]. In this work we obtain the missing contributions $A^{(2,1)}$ and $A^{(2,\tilde{N}_f)}$. Representative diagrams for each contribution are shown in figure 1.

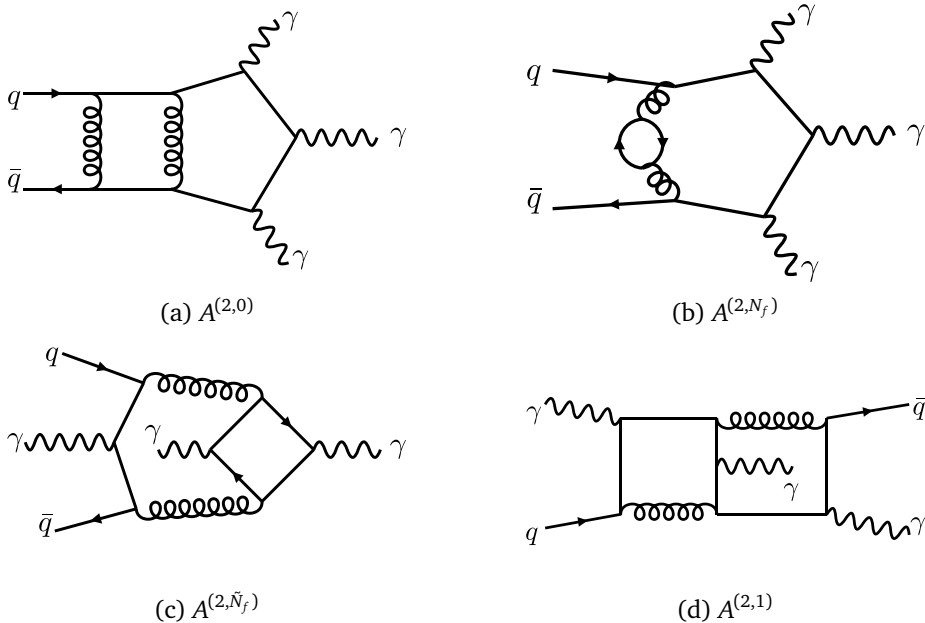

(a) $A^{(2,0)}$

(b) $A^{(2,N_f)}$

(c) $A^{(2,\tilde{N}_f)}$

(d) $A^{(2,1)}$

Figure 1: Representative diagrams for the different contributions to eq. (14).

The renormalised amplitudes still contain infrared (IR) poles. They can be predicted and subtracted in a scheme-dependent way. We use the scheme of refs. [51, 52, 54], which we denote as the Catani scheme. The finite remainders $\mathcal{R}$ are obtained from the UV-renormalised amplitudes $\mathcal{A}$ through

$$\mathcal{R} := \mathbf{I}\mathcal{A}. \tag{17}$$

Upon expansion of eq. (17) in $\alpha_s$ and $\epsilon$, we obtain

$$
\begin{aligned}
\mathcal{R}^{(0)} &= \mathcal{A}^{(0)}, \\
\mathcal{R}^{(1)} &= \mathcal{A}^{(1)} + \mathbf{I}^{(1)}\mathcal{A}^{(0)} + \mathcal{O}(\epsilon), \\
\mathcal{R}^{(2)} &= \mathcal{A}^{(2)} + \mathbf{I}^{(1)}\mathcal{A}^{(1)} + \mathbf{I}^{(2)}\mathcal{A}^{(0)} + \mathcal{O}(\epsilon),
\end{aligned}
\tag{18}
$$

where the functions $\mathbf{I}^{(1)}$, $\mathbf{I}^{(2)}$ are the coefficients of the expansion of $\mathbf{I}$ in $\alpha_s/(2\pi)$, and are given in appendix A.1.[1] Up to two-loops we can expand the finite remainders into the gauge-invariant contributions $R^{(1)}$, $R^{(2,0)}$, $R^{(2,1)}$, $R^{(2,N_f)}$ and $R^{(2,\tilde{N}_f)}$ as in eqs. (14) and (15).

For phenomenological applications, we are mostly interested in squared finite remainders, summed over helicity and color, which we will refer to as the *hard function*. We define it as

$$\mathcal{H} = \frac{1}{\mathcal{B}} \sum_h |\mathcal{R}_h|^2, \qquad \mathcal{B} := \sum_h \left|\mathcal{A}_h^{(0)}\right|^2, \tag{19}$$

where the sum is over all different choices of helicities of the involved particles. Expanding in $\alpha_s$ and in color factors, we get

$$
\begin{aligned}
\mathcal{H}^{(1)} &= C_F H^{(1)}, \\
\mathcal{H}^{(2)} &= \frac{N_c^2}{4}\left(H^{(2,0)} - \frac{1}{N_c^2}(H^{(2,0)} + H^{(2,1)}) + \frac{1}{N_c^4}H^{(2,1)}\right) \\
&\quad + C_F T_F N_f H^{(2,N_f)} + C_F T_F \left(\sum_{f=1}^{N_f} Q_f^2\right) H^{(2,\tilde{N}_f)},
\end{aligned}
\tag{20}
$$

---

[1] We note that our definition of $\mathbf{I}^{(1)}$, $\mathbf{I}^{(2)}$ differs by a sign from the one of ref. [48].

and by definition $\mathcal{H}^{(0)} = 1$. We note that the hard functions are scheme dependent, with expressions in different schemes being related by a finite shift. We give examples of such relations in appendix A.2.

# 3 Calculation

## 3.1 Overview

Our calculation is done within the framework of two-loop numerical unitarity [27–29]. We rely on the implementation of the method in the program CARAVEL [31].

This approach starts from the observation that the integrands of the gauge-invariant contributions $A^{(2,k)}$ of eq. (15) all admit a decomposition of the form

$$A(\ell) = \sum_{\Gamma} \sum_{i \in M_\Gamma \cup S_\Gamma} c_{\Gamma,i} \frac{m_{\Gamma,i}(\ell)}{\prod_j \varrho_{\Gamma,j}(\ell)} \,, \tag{21}$$

where the outer sum is over all distinct sets $\Gamma$ of inverse propagators $\varrho_{\Gamma,j}$ contributing to the amplitude (which we call *topologies*). For each topology $\Gamma$, the numerators $m_{\Gamma,i}(\ell)$ are polynomials in the loop momenta $\ell$ (we use a single $\ell$ to collectively denote all loop momenta), and they are constructed such that each $m_{\Gamma,i}(\ell)$ either corresponds to a master integral ($i \in M_\Gamma$) or can be expressed as a total derivative and therefore integrates to zero, i.e. it is a surface term ($i \in S_\Gamma$). The coefficients $c_{\Gamma,i}$ in eq. (21) are unknown rational functions of the external particles' momenta and the dimensional regulator $\epsilon$. In the generalized unitarity method, the coefficients of each topology $\Gamma$ are constrained by evaluating (generalized) *cuts*, corresponding to residues of eq. (21) at values of $\ell = \hat{\ell}$ such that $\varrho_{\Gamma,j}(\hat{\ell}) = 0 \ \forall j$. The cuts of the LHS of eq. (21) are evaluated as products of $D$-dimensional tree-amplitudes (which we call *cut diagrams*) and matched to the integrand parametrisation of eq. (21). In this way, *cut equations* are generated for various loop-momentum configurations $\hat{\ell}_k$ satisfying the cut conditions $\varrho_{\Gamma,j}(\hat{\ell}_k) = 0$. Given a large enough sample of $\hat{\ell}_k$, the coefficients $c_{\Gamma,i}$ in eq. (21) can be determined as (numerical) solutions to the cut equations. Our computation is based on the public implementation of the two-loop numerical unitarity method in CARAVEL [31], which we extended to account for the non-planar topologies appearing in eq. (21). These were absent in previous applications of the two-loop numerical unitarity approach. In particular, we constructed the complete set of surface terms required for the non-planar contributions in $A^{(2,1)}$ and $A^{(2,\tilde{N}_f)}$. We discuss this in more detail in section 3.2.

We note that to be compatible with generalized cuts in $D$ dimensions, we require that the numerators $m_{\Gamma,i}(\ell)$ in eq. (21) are polynomials in the loop momenta (i.e. additional denominator powers are not allowed). In addition, for each $\Gamma$ the numerators $m_{\Gamma,i}(\ell)$ must be linearly-independent on $D$-dimensional cuts that set all $\varrho_{\Gamma,j}$ to zero. While for the surface terms these condition hold by construction, the definitions of master integrals are frequently chosen such that at least one of the conditions is violated. Indeed, the basis of master integrals of refs. [55, 56] violates the second condition. We find that the basis of ref. [57] is therefore more convenient for our approach. It can easily be written in terms of the pentagon functions of ref. [56] using modern integration-by-parts codes, e.g. [58, 59], which among other benefits allows for an efficient numerical evaluation of the master integrals. The decomposition of eq. (21) then leads to a decomposition of the finite remainders in eq. (18) in terms of pentagon functions,

$$\mathcal{R}^{(\ell)} = \sum_i r_i h_i \,, \tag{22}$$

where the $r_i$ are rational functions of external kinematics, and the $h_i$ are monomials of the

pentagon functions of ref. [56]. In summary, two-loop numerical unitarity gives us a way to numerically compute the $r_i$. We can then use these numerical evaluations to reconstruct their analytic form.

Before discussing some details of the construction of surface terms in section 3.2, and of the analytic reconstruction of the finite remainders in section 3.3, we close this brief overview of the approach with some technical comments. Cut diagrams are generated with `qgraf` [60], and arranged into a hierarchy of cuts with a private code. To match the cuts evaluated through color-ordered tree amplitudes to the amplitude definitions in section 2 we employ the unitarity-based colour decomposition of refs. [61, 62]. We determine the $\epsilon$-dependence of cut diagrams originating from the state sums in the loops through the dimensional reduction method developed in refs. [63–65]. This allows us to perform the entire calculation with six-dimensional states only.

## 3.2 Surface Terms

The two-loop numerical unitarity framework [27–29] builds on the parametrisation of the integrand as in eq. (21). A crucial step in this procedure is the determination of a basis of surface terms that integrate to zero. In this section we present the method we used to construct surface terms for the non-planar topologies.

The construction of surface terms starts from the observation that total derivatives of Feynman integrands integrate to zero in dimensional regularisation [39, 40, 66]. That is, we can construct surface terms from

$$\int \mathrm{d}^D \ell_1 \cdots \mathrm{d}^D \ell_L \frac{\partial}{\partial \ell_a^\mu} \frac{v_a^\mu(\ell)}{\varrho_1 \cdots \varrho_N} = 0 \,. \tag{23}$$

For an arbitrary vector $v_a^\mu$, the equation above will generate surface terms involving integrals that are not relevant for the amplitudes we are computing, either because they have numerators of too high degree or because they have propagators raised to too high powers. In order to have an efficient construction of the surface terms required for the decomposition in eq. (21), it is beneficial to construct a minimal set of surface terms, which contain specific propagator powers and numerators whose polynomial degree is limited by the interactions of the underlying process.

While the numerator power in the surface terms is easy to control (because derivatives do not increase it), care must be taken with the power of the propagators. Propagator powers in surface terms can be controlled by requiring that the vector $v_a^\mu$ in eq. (23) satisfies [41]

$$\sum_{a,\mu} v_a^\mu(\ell) \frac{\partial \varrho_i}{\partial \ell_a^\mu} = f_i(\ell) \varrho_i \,, \quad \forall i \,. \tag{24}$$

where the unknowns $f_i$ and $v_a^\mu$ are polynomials in loop momenta. We call such vectors unitarity-compatible integration-by-parts (IBP) generating vectors, or simply IBP-generating vectors. The mathematical structure of eq. (24) is well known and defines the vectors $v_a^\mu$ and the $f_i$ to be elements of a syzygy module.

Solving eq. (24) allows us to construct a minimal set of surface terms. In practice, however, obtaining analytic expressions for the surface terms can be challenging, both due to the difficulty of solving eq. (24) and due to the size of the final expressions (see refs. [67–72] for related work). Since our goal is to reconstruct analytic expressions for the two-loop remainders from their numerical evaluations on a sufficient number of phase-space points, we do not actually require analytic solutions to eq. (24). Indeed, it is sufficient to obtain IBP-vectors that are analytic in the loop-momentum variables but numerical in external momenta. Naturally, this requires solving eq. (24) at each phase-space point, and one must therefore have

an approach that is efficient. We now present our solution to this problem: we first discuss its formulation in embedding space [43, 73], and then discuss how unitarity-compatible IBP vectors and surface terms are constructed at a given phase-space point, and how we efficiently extend this to subsequent phase-space points.

### 3.2.1  Embedding-Space Formalism

We start by reviewing the formulation of Feynman integrals in embedding space. We will observe in the next section that this formulation simplifies the solution of the syzygy equations in eq. (24). Formally, our goal is to discuss how momentum space can be mapped into a subset of a projective space, commonly called embedding space. To this end, we map each point $z^\mu$ in momentum space into a line $Z$ of projectively equivalent points. In the context of a Feynman integral with $N$ external legs, this can be done as follows [43, 73]. We first define

$$q_i^\mu = \sum_{j=1}^{i-1} k_j^\mu, \qquad 1 \le i \le N, \tag{25}$$

where the $k_j^\mu$ denote the external momenta and the empty sum gives the zero vector. We then arrange the loop and external momenta into the $(D+2)$-dimensional embedding-space vectors

$$Y_a = c_{Y_a} \left( \ell_a^\mu, (\ell_a)^2, 1 \right), \quad X_i = c_{X_i} \left( q_i^\mu, (q_i)^2, 1 \right), \tag{26}$$

where $c_{Y_a}$ and $c_{X_i}$ parametrise the points in embedding space that are projectively equivalent. As a projective space, embedding space is equipped with a special point, commonly referred to as the infinity point and denoted here by $X_0$,

$$X_0 = (0^\mu, 1, 0). \tag{27}$$

We also define an inner product as

$$(AB) = c_A c_B \left( -2a^\mu \cdot b_\mu + a^2 + b^2 \right). \tag{28}$$

An $L$-loop Feynman integral is then mapped into embedding space as

$$\int \left[ \prod_{a=1}^{L} \mathrm{d}^D \ell_a \right] f(\ell; q) = \int \left[ \prod_{a=1}^{L} \frac{d^{D+2} Y_a}{\mathrm{vol}[\mathrm{GL}(1)]} \frac{\delta[(Y_a Y_a)]}{(X_0 Y_a)^D} \right] F(Y; X), \tag{29}$$

where $F(Y; X)$ is implicitly defined as the image of $f(\ell; q)$ under the embedding procedure and $\mathrm{vol}[\mathrm{GL}(1)]$ accounts for the projective equivalence parametrised by the $c_{Y_a}$.

In this construction, we seem to have increased the number of degrees of freedom from $D$ to $D+2$. One apparent extra degree of freedom is removed by noting that any $z^\mu$ in momentum space is mapped to an embedding space point $Z$ that satisfies $(ZZ) = 0$, i.e., onto the light-cone in embedding space. Indeed, we can easily check from eq. (26) that $(Y_a Y_a) = 0$ and $(X_i X_i) = 0$. The integration in embedding space in eq. (29) is restricted to the light-cone by the delta function $\delta[(Y_a Y_a)]$, thereby corresponding to the integration over all loop-momentum space. Another apparent extra degree of freedom is parametrised by the non-vanishing parameters $c_{Y_a}$ and $c_{X_i}$. To understand how it is removed in practice, we first note that, for an arbitrary embedding-space vector $Z$, $c_Z = (X_0 Z)$. It therefore follows that

$$(\ell_1 - \ell_2)^2 = \frac{(Y_1 Y_2)}{(X_0 Y_1)(X_0 Y_2)}, \quad (\ell_a - q_i)^2 = \frac{(Y_a X_i)}{(X_0 Y_a)(X_0 X_i)}, \quad (q_i - q_j)^2 = \frac{(X_i X_j)}{(X_0 X_i)(X_0 X_j)}. \tag{30}$$

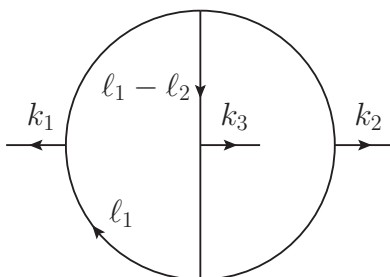

Figure 2: Non-planar diagram corresponding to the propagators in eq. (34).

The function $f(\ell; q)$ in eq. (29) is a rational function of these momentum-space inner products, which implies that $F(Y; X)$ is homogeneous of degree 0 in both $c_{Y_a}$ and $c_{X_i}$, that is

$$F(Y; X) = F(Y_1, \ldots, \lambda Y_a, \ldots, Y_L; X) = F(Y; X_1, \ldots, \lambda X_i, \ldots, X_N), \qquad \lambda \neq 0. \tag{31}$$

Since the $X_i$ only appear in eq. (29) within $F(Y; X)$, we can set $c_{X_i} = (X_0 X_i) = 1$. In principle we could make the same choice for $c_{Y_a}$, but this would obscure some of the properties of the Feynman integral, such as the fact that in dimensional regularization the integrand has a branch point at $(X_0 Y_a) = 0$ [43, 45, 73]. Equivalently, we can keep the explicit dependence on the $(X_0 Y_a)$, and the division by vol[GL(1)] ensures that the integral is well defined despite the invariance of $F(Y; X)$ with respect to rescalings of each $Y_a$.

The (inverse) propagators in a Feynman integral can all be written in terms of the expressions given in eq. (30), where we are free to set $(X_0 X_i) = 1$ as discussed above. For two-loop five-point non-planar integrals it is also convenient to define another combination of the form

$$(\ell_1 - \ell_2 + k_i)^2 = \frac{r_{AB} Y_1^A Y_2^B}{(X_0 Y_1)(X_0 Y_2)}, \tag{32}$$

where $r_{AB}$ is a matrix that depends only on external kinematics. That is, the components of $r_{AB}$ can be written in terms of the $(X_i X_j)$ and are independent of $Y_1$ and $Y_2$. The denominator $(\ell_1 - \ell_2)^2$ is a particular case of eq. (32). We denote the set of propagator denominators in a two-loop five-point non-planar integral as

$$\mathcal{P} = \left\{ (X_1 Y_1), (X_2 Y_1), \ldots, (Y_1 Y_2), r_{AB} Y_1^A Y_2^B \right\}, \tag{33}$$

where we note that the last term is not required for planar integrals.

For concreteness we explicitly give an example of a simple non-planar diagram in embedding space. The propagators of the diagram in fig. 2 take the form

$$\varrho_1 = \ell_1^2 = \frac{(X_1 Y_1)}{(X_0 Y_1)}, \quad \varrho_2 = (\ell_1 - k_1)^2 = \frac{(X_2 Y_1)}{(X_0 Y_1)}, \quad \varrho_3 = (\ell_2 + k_3)^2 = \frac{(X_3 Y_2)}{(X_0 Y_2)},$$

$$\varrho_4 = (\ell_2 + k_2 + k_3)^2 = \frac{(X_2 Y_2)}{(X_0 Y_2)}, \quad \varrho_5 = (\ell_1 - \ell_2)^2 = \frac{(Y_1 Y_2)}{(X_0 Y_1)(X_0 Y_2)}, \tag{34}$$

$$\varrho_6 = (\ell_1 - \ell_2 - k_3)^2 = \frac{1}{(X_0 Y_1)(X_0 Y_2)} [(Y_1 Y_2) + (X_0 Y_1)(X_3 Y_2) -$$

$$(X_0 Y_1)(X_1 Y_2) + (X_0 Y_2)(X_1 Y_1) - (X_0 Y_2)(X_3 Y_1) + (X_0 Y_1)(X_0 Y_2)(X_1 X_3)],$$

where $\varrho_6$ has the form of eq. (32) and we have written the components of the respective matrix $r_{AB}$ explicitly in terms of the $(X_i X_j)$.

### 3.2.2  IBP-Generating Vectors In Embedding Space

Let us now discuss how IBP relations arise in embedding space, following previous work on the subject [43, 44]. To begin, we reformulate eq. (23) in embedding space as[2]

$$
\int \prod_a \left[ \frac{d^{D+2} Y_a}{\text{vol}[\text{GL}(1)]} \right] \times \frac{\partial}{\partial Y_c^C} \left[ \frac{V_c^C}{\varrho_1 \cdots \varrho_N} \prod_b \frac{\delta[(Y_b Y_b)]}{(X_0 Y_b)^D} \right] = 0 \,,
\tag{35}
$$

where the $\rho_i$ are now understood as their embedding-space expressions. The IBP-generating vectors are denoted $V_c^C$, where $C$ runs over the $D + 2$ dimensions of an embedding-space vector. A vector $V_c$ that generates linear relations between Feynman integrals in embedding space must satisfy a number of non-trivial properties. First, the components of $V_c^C$ must be homogeneous functions in $Y_c$ of degree one in order to compensate for the scaling of the partial derivatives. More precisely, they are required to be rational functions with denominators given by products of the factors $(X_0 Y_a)$, so that no further propagator poles are introduced. Second, it can be verified that

$$
\frac{\partial}{\partial Y_c^C} \left[ \frac{F_c Y_c^C}{\varrho_1 \cdots \varrho_N} \prod_b \frac{\delta[(Y_b Y_b)]}{(X_0 Y_b)^D} \right] = 0 \,,
\tag{36}
$$

for any homogeneous $F_c$ that is degree 0 in $Y_c$. As such, any IBP-generating vector $V_c$ generates the same surface term as another vector obtained from it by translations in the $Y_c$ direction. Finally, any valid IBP vector must only generate relations between Feynman integrals. We therefore impose that any terms containing derivatives of the delta function that arise from expanding the derivative in eq. (35) should cancel. This can be achieved by requiring that

$$
V_c^C \frac{\partial}{\partial Y_c^C} (Y_b Y_b) = F_b^{\text{quadric}} (Y_b Y_b), \qquad \forall b \,,
\tag{37}
$$

i.e. by requiring that the $V_c$ generate translations along the light cone. It turns out that many of the solutions to eq. (37) can be discarded as they lead to vanishing surface terms. Indeed, it is easy to see that a subset of the solutions is given by $V_c$ such that $V_c = \frac{1}{2} F_c^{\text{quadric}} Y_c$. By comparison with eq. (36), we see that such solutions give a vanishing surface term. We can use this observation to simplify IBP-vector construction. Specifically, if we have an IBP vector $V_c$ for which the associated $F_c^{\text{quadric}} \neq 0$, then the same surface term is generated by a vector $V_c' = V_c - \frac{1}{2} F_c^{\text{quadric}} Y_c$. It follows that the new vector $V_c'$ satisfies equation (37) with $F_b^{\text{quadric}} = 0$. Therefore, the solutions of eq. (37) with $F_b^{\text{quadric}} = 0$ generate the full set of surface terms and it is sufficient to require that the IBP vectors $V_c$ satisfy

$$
V_c^C \frac{\partial}{\partial Y_c^C} (Y_b Y_b) = 0, \qquad \forall b \,.
\tag{38}
$$

Within the two-loop generalised unitarity framework, we also require that IBP-generating vectors are unitarity compatible, that is that they satisfy eq. (24). In embedding space, the analogous condition is that the exponents of the denominator factors $(X_i Y_a)$ and $\left( r_{AB} Y_1^A Y_2^B \right)$ are not increased [44]. For propagators that depend on a single $Y_a$, the conditions read,

$$
V_c^C \frac{\partial}{\partial Y_c^C} (X_i Y_b) = F_i (X_i Y_b) \,,
\tag{39}
$$

---

[2]In practice, it can be more convenient to replace the $\delta[(Y_b Y_b)]$ by $1/(Y_b Y_b)$ and modify the integration contour to encircle the pole at $(Y_b Y_b)$. This perspective can be taken to more easily derive some of the properties of IBP vectors in embedding space.

and for the propagators that depend on both loop momenta

$$V_c^C \frac{\partial}{\partial Y_c^C} \left( r_{AB} Y_1^A Y_2^B \right) = F_r \left( r_{AB} Y_1^A Y_2^B \right), \tag{40}$$

which also covers the case of the propagator $(Y_1 Y_2)$.

In summary, we must construct IBP-generating vectors satisfying eqs. (38) to (40). These constrain the vectors $V_a^A$ and the polynomials $F_i$ and $F_r$ to be elements of the syzygies of a module defined by

$$\begin{aligned}
(V_a Y_a) &= 0, \quad \forall a, \\
(V_a X_i) &= F_i (X_i Y_a), \quad \text{for } (X_i Y_a) \in \mathcal{P}, \\
r_{AB} \left[ V_1^A Y_2^B + V_2^B Y_1^A \right] &= F_r \left( r_{AB} Y_1^A Y_2^B \right), \quad \text{for } (Y_1 Y_2) \text{ and } \left( r_{AB} Y_1^A Y_2^B \right) \in \mathcal{P},
\end{aligned} \tag{41}$$

where $\mathcal{P}$ was defined in eq. (33). The unknowns in these equations are the $V_a^A$, $F_i$ and $F_r$. From now on, for simplicity we set $(X_0 Y_a) = 1$ and require the solutions to be polynomials in the $(X_i Y_a)$ and $(Y_1 Y_2)$, keeping in mind that those factors can be reinstated by requiring that quantities have the correct homogeneous degree.

### 3.2.3 Computing IBP-Generating Vectors

We now turn towards solving the syzygy equations (41). In short, our strategy is to construct a degree-bounded generating set of solutions on a single phase-space point and then use the shape of this set to compute a generating set of vectors on further phase space points, via linear algebra. We start by expressing the IBP vectors in terms of the momenta in the problem,

$$V_a = \sum_{i=0}^{N} v_a^i X_i + v_a^Y Y_{b \neq a}, \tag{42}$$

where we took into consideration eq. (36) and the surrounding discussion to remove the $Y_a$ direction. We can then insert this parametrisation into eq. (41) and re-express these equations as

$$M_{IJ} w^J = 0, \quad w^J = \{ v_1^i, v_1^Y, v_2^i, v_2^Y, F_i, F_r \}, \tag{43}$$

where the vector $w^J$ collects all unknowns in the problem (while it contains a single $v_1^Y$ and $v_2^Y$, there are several $v_1^i$ and $v_2^i$, several $F_i$ and two $F_r$). The direct solution of eq. (43) is in general very challenging. Since we are only interested in a subset of solutions that generates sufficiently many surface terms to construct the decomposition in eq. (21), which is degree bounded by the theory, we instead solve the problem for a fixed polynomial degree [29,67,72]. To make this degree manifest, we consider the ansatz

$$w^J = \sum_{|\vec{m}| \leq Q} w_{\vec{m}}^J y^{\vec{m}}, \qquad y = \{ (X_1 Y_1), \dots, (X_1 Y_2), \dots, (Y_1 Y_2) \}, \tag{44}$$

where the $w_{\vec{m}}^J$ are $Y$-independent rational functions of the external kinematics, i.e. of the $(X_i X_j)$, that multiply associated monomials of the $y_k$ variables,

$$y^{\vec{m}} := \prod_k y_k^{m_k}, \tag{45}$$

where the multi-index $\vec{m}$ specifies the exponents of the monomial. The total degree of a monomial $y^{\vec{m}}$ is denoted by $|\vec{m}|$,

$$|\vec{m}| := \sum_k m_k. \tag{46}$$

Inserting eq. (44) into eq. (43) one obtains a linear system for the coefficients $w_{\vec{m}}^J$ allowing us to solve the syzygy problem with a degree bound $Q$ as a linear algebra problem.

We can however further simplify the problem. Indeed, we note that the matrix $M_{IJ}$ is linear in the $y_k$, that is

$$M_{IJ} = M_{IJ}^{[0]} + M_{IJ}^{[1]}, \qquad M_{IJ}^{[1]} = \sum_k M_{IJk}^{[1]} y_k. \tag{47}$$

It is interesting to note that, as the $X_i$ are linearly independent, $M_{IJ}^{[1]}$ is independent of external kinematics, i.e., the $M_{IJk}^{[1]}$ are matrices of rational numbers. We can also decompose the ansatz of eq. (44) into leading and subleading $y$ contributions,

$$w^J = w_{\text{max}}^J + w_{\text{rem}}^J, \qquad w_{\text{max}}^J = \sum_{|\vec{m}|=Q} w_J^{\vec{m}} y^{\vec{m}}. \tag{48}$$

It then follows that

$$M_{IJ}^{[1]} w_{\text{max}}^J = 0, \tag{49}$$

which is a simpler syzygy problem than eq. (43) as it is homogeneous and independent of kinematics. For $w^J$ to be a solution to eq. (43), it is necessary that its maximal degree piece solves eq. (49).

To solve eq. (49) we use the methods implemented in SINGULAR [74], which yield a generating set of solutions, with each generator homogeneous of a given degree. That is, we have the generating set,

$$\left\{ w_k^{J,[q_k]}(\vec{y}) \right\}_{k=1,\dots k_{\text{max}}}, \tag{50}$$

where $q_k$ is the degree of the homogeneous generator $w_k^{J,[q_k]}$. A basis of solutions with degree $q$ to eq. (49) is obtained from linear combinations of the generating set of eq. (50) with coefficients that are homogeneous polynomials of degree $q - q_k$.

We now use the homogeneous syzygies of eq. (49) to construct solutions to the full syzygy problem of eq. (43). Since we require solutions only up to a maximal polynomial degree we again consider solutions up to a given polynomial degree $Q$. To this end, we consider a parametrization of a syzygy with degree $q$ that consists of

1. a maximal degree piece of degree $q$ constructed from the generators eq. (50). This consists of combinations of the generators multiplied with polynomials of suitable polynomial degree, and,

2. a generic polynomial-valued vector with degree less than $q$.

More precisely, such a syzygy can be written as

$$v^{I,[q]}(\vec{y}) = \sum_{k,\, |\vec{m}_k|+q_k=q} b_{k,\vec{m}_k}\, y^{\vec{m}_k}\, w_k^{I,[q_k]}(\vec{y}) + \sum_{|\vec{m}|<q} b_{\vec{m}}^I\, y^{\vec{m}}, \tag{51}$$

for $y$-independent coefficients $b_{k,\vec{m}_k}$ and $b_{\vec{m}}^I$. The monomials $y^{\vec{m}}$ and $y^{\vec{m}_k}$ are bounded by the degree constraints $|\vec{m}| < q$ and $|\vec{m}_k| < q$. Inserting this ansatz into eq. (43) then yields linear equations for $b_{k,\vec{m}_k}$ and $b_{\vec{m}}^I$. In practice, we observe that this approach of combining maximal degree solutions with a parametrization of lower degree terms significantly reduces the size of the linear system compared to starting from the generic parametrisation of eq. (44). In this way, working degree by degree we can construct a basis of the solutions up to degree $Q$.

Once we have constructed this basis of the solutions it is easy to find a subset of the basis elements which generate the module up to degree $Q$. Working degree by degree, one simply removes basis elements that are polynomial combinations of lower degree generators using linear algebra.

At this point we have obtained a generating set of the solutions of eq. (43) up to polynomial degree $Q$ at a given numerical phase-space point. We now use this information to streamline the construction of IBP vectors on further phase-space points. We start with the generating set we have obtained and express their components as a linear combination of monomials

$$v^I(\vec{y}) = \sum_{\vec{m}} r^I_{\vec{m}} y^{\vec{m}}, \tag{52}$$

where we sum over all monomials such that the associated $r^I_{\vec{m}}$ are non-zero. We then replace the numerical coefficients $r^I_{\vec{m}}$ by parameters $b^I_{\vec{m}}(\vec{p})$ that explicitly depend on the external kinematic point $\vec{p}$, in order to obtain *skeleton* vectors

$$v^I_{\text{skel}}(\vec{y}) = \sum_{\vec{m}} b^I_{\vec{m}}(\vec{p}) \vec{y}^{\vec{m}}. \tag{53}$$

We finally insert these skeleton vectors into (43), and obtain a linear system of equations that constrain the $b^I_{\vec{m}}(\vec{p})$. Importantly, these linear systems are analytic in the external kinematics and so can be solved phase-space point by phase-space point. These equations may be linearly dependent and not uniquely determine the $b^I_{\vec{m}}(\vec{p})$. We discard linearly dependent equations, and choose a set of $b^I_{\vec{m}}(\vec{p})$ to set to zero so that the solution is unique. By making use of these skeleton vectors instead of fully general vectors, we obtain equations for a minimal set of monomial coefficients. In practice, the largest linear systems that we encounter when solving for the $b^I_{\vec{m}}$ of eq. (53) are approximately $2000 \times 2000$. Finally, we note that for simpler case we find it efficient to solve these linear systems analytically.

### 3.2.4 Computing Surface Terms

Now that we have constructed a set of IBP generating vectors up to degree $Q$, we use them to construct a collection of surface terms. Recalling eq. (21), our goal is to construct a set of linearly-independent surface terms $S_\Gamma$ for a given topology $\Gamma$ that is sufficiently large for the power counting of our theory. This set of surface terms is a subset of the $m_{\Gamma,i}$ of eq. (21) and they are polynomials in the loop momentum components. The maximum degree of these polynomials is known a priori, since it is determined by the theory describing the scattering process, and as such it is easy to construct a basis of span$(M_\Gamma \cup S_\Gamma)$. As a basis of master integrals is also known [55–57], we have sufficient information to construct a linearly-independent set of surface terms.

To construct a single surface term, we start with an IBP-generating vector and a monomial of the $y_k$ variables. We take their product, and use this as the vector $V^C_c$ in eq. (23). Expanding out the derivative leads to a surface term in embedding space that is easily mapped to one in momentum space by identifying all $y_k$ variables with their momentum space counterparts. In this way, taking all pairs of IBP-vectors and monomials of $y_k$ variables that satisfy our power counting bounds, we construct a set of surface terms. Many of the surface terms obtained in this way are linearly dependent and can be discarded. As our construction of the IBP-generating vectors was only performed up to degree $Q$, the set of surface terms generated in this way may not be a basis of span$(S_\Gamma)$. In practice, to solve this problem, we repeat the procedure described in section 3.2.3, increasing the value of $Q$ in order to construct more IBP generating vectors until our set of surface terms is a basis of span$(S_\Gamma)$.

### 3.3 Analytic Reconstruction In Spinor Helicity Formalism

As noted above, the two-loop numerical unitarity approach allows us to numerically evaluate the coefficients $r_i$ in the finite remainder of eq. (22). We now discuss how to obtain analytic expressions using these numerical evaluations. As suggested in e.g. refs. [36,75], we will consider the $r_i$ to be rational functions of the spinor-helicity variables $\lambda$ and $\tilde{\lambda}$ already introduced in section 2.[3] The coefficients admit a least common denominator representation, which reads

$$r_i(\lambda, \tilde{\lambda}) = \frac{\mathcal{N}_i(\lambda, \tilde{\lambda})}{\prod_j \mathcal{D}_j^{q_{ij}}(\lambda, \tilde{\lambda})}. \tag{54}$$

The exponents $q_{ij}$ are allowed to take negative values, thus denoting numerator factors. The $\mathcal{N}_i$ and $\mathcal{D}_j$ are polynomials in spinor brackets. Like the amplitude, the $r_i$ are dimensionful and transform non-trivially under the little group. For a spinor function $\mathcal{E}(\lambda, \tilde{\lambda})$ we define the mass dimension $[\mathcal{E}]$ and $k^{\text{th}}$ little-group weight $\{\mathcal{E}\}_k$ through

$$\mathcal{E}(z\lambda_1, \ldots, z\lambda_n, z\tilde{\lambda}_1, \ldots, z\tilde{\lambda}_n) = z^{2[\mathcal{E}]} \mathcal{E}(\lambda_1, \ldots, \lambda_n, \tilde{\lambda}_1, \ldots, \tilde{\lambda}_n), \tag{55}$$

$$\mathcal{E}(\ldots, z\lambda_k, \ldots, \tilde{\lambda}_k/z, \ldots) = z^{\{\mathcal{E}\}_k} \mathcal{E}(\ldots, \lambda_k, \ldots, \tilde{\lambda}_k, \ldots). \tag{56}$$

The mass dimension of an $n$-point amplitude $\mathcal{A}_n$ is $[\mathcal{A}_n] = 4 - n$. The $k^{\text{th}}$ little-group weight of an amplitude depends on the $k^{\text{th}}$-particles' helicity $h_k$,

$$\{\mathcal{A}\}_k = -2h_k. \tag{57}$$

Crucially, there is by now a large collection of evidence that the $\mathcal{D}_j$ in eq. (54) are known a priori. Indeed, they can be constructed from the symbol alphabet [30]. In our normalisation of the amplitudes (see eq. (8)), they also include factors with little-group weight, such as the tree level amplitude if it is not zero, or the leading order of the one-loop amplitude. Given that the pentagon functions $h_i$ in eq. (22) are dimensionless and little-group invariant, the mass dimensions and little-group weights of the $r_i$ are the same as those of the amplitudes. Since the $\mathcal{D}_j$ are known, the mass dimensions and little-group weights of the numerators $\mathcal{N}_i$ are easily determined.

In a nutshell, the reconstruction procedure amounts to constructing and fitting an ansatz for the $\mathcal{N}_i$ in terms of spinor variables. An important feature of spinor-helicity-based reconstruction methods is that the ansatz directly captures the little-group and mass-dimension properties of the coefficients. This is in contrast to Mandelstam-based reconstruction methods, where the $r_i$ are normalized with a (dimensionful) phase factor to be little-group invariant, and split into parity even and odd parts. This difference has an important practical consequence, as the common denominator form of the $r_i$ often simplifies in spinor variables. For example, the symbol alphabet may factorize in spinor variables. This can lead to denominator factors that cancel further against the coefficient's numerators, lowering the mass dimensions of the latter, which thus require simpler ansätze in the reconstruction procedure.

To begin constructing our ansatz, we specify the set of denominator factors $\mathcal{D}_j$ in eq. (54). As already alluded to, these denominator factors can be constructed from the symbol alphabet. In Mandelstam variables they correspond to the subset of so-called even letters, which describe the collection of singular surfaces associated to the master integrals. When written in terms of spinor variables, these surfaces may branch. That is, alphabet letters that appear irreducible when written in terms of Mandelstam variables may further factorize in spinor space. Beyond alphabet letters, the $\mathcal{D}_j$ also include rational functions appearing in tree-level

---

[3]More precisely, the coefficients belong to the field of fractions of the ring of independent Lorentz invariants; see ref. [36, Section 2.2] for more details.

or one-loop amplitudes. We therefore take these functions and the irreducible factors of the symbol alphabet in spinor space as the set of expected denominator factors. This analysis was already performed in ref. [36] for the amplitudes considered in this paper, and it was found that the expected set of denominator factors contains 35 elements. It can be expressed as

$$\mathcal{D} = \left\{ \langle ij \rangle, [ij] \, : \, 1 \leq i < j \leq 5 \right\} \cup \left( \bigcup_{\sigma \in Z_5} \sigma \circ \left\{ \langle 1|2+3|1], \langle 1|2+4|1], \langle 1|2+5|1] \right\} \right), \quad (58)$$

where the elements of $Z_5$ are the cyclic permutations acting on the momentum indices associated to each spinor. Note that, despite its presence in the alphabet we do not include $\text{tr}_5$ in $\mathcal{D}$, as it is expected to cancel in the finite remainder (see e.g. refs. [57, 76, 77]).

In the following, it will be useful to consider the zero set associated to various denominators. We denote the algebraic variety corresponding to the common zero set associated to a list of denominator factors $\{\mathcal{D}_{i_1}, \ldots, \mathcal{D}_{i_n}\}$ by

$$V(\mathcal{D}_{i_1}, \ldots, \mathcal{D}_{i_n}). \quad (59)$$

**Denominator exponents**    Our first task is to determine the denominator exponents $q_{ij}$ in eq. (54). We will use the standard technology of univariate-slice reconstruction [30], combined with the approach introduced in ref. [78], based on all-line BCFW-shifts [79–81]. Its application to functions of Mandelstam variables was discussed in ref. [47]. Here we review the procedure and discuss its generalization to functions of spinor variables.

In order to determine the $q_{ij}$, we must choose a set of curves in phase space that intersect each surface $V(\mathcal{D}_j)$ at a generic point at least once and are not contained in any of the $V(\mathcal{D}_j)$. We will achieve this with two BCFW shifts, one holomorphic and one anti-holomorphic, following the all-line shift approach of ref. [47] which we now review. Let us begin with a generic, numeric momentum-conserving configuration of spinors $\{\lambda_1, \tilde{\lambda}_1, ..., \lambda_5, \tilde{\lambda}_5\}$. We make an all-line holomorphic shift by adjusting every $\lambda$ spinor in a way which is proportional to a common reference spinor $\eta$. That is we shift

$$\lambda_i \to \lambda_i + t c_i \eta, \quad \tilde{\lambda}_i \to \tilde{\lambda}_i. \quad (60)$$

Here, we introduce unknowns $c_i$ that we use to ensure that the shifted kinematics satisfy momentum conservation. Specifically, we choose the $c_i$ to satisfy

$$\sum_{i=1}^{5} c_i \tilde{\lambda}_i = 0. \quad (61)$$

This linear equation does not have a unique solution. Nevertheless, any solution with $c_i \neq 0$ will guarantee a non-trivial shift that satisfies momentum conservation. In order to ensure that a slice is generic, we pick both the initial set of spinor variables $\{\lambda_1, \tilde{\lambda}_1, ..., \lambda_5, \tilde{\lambda}_5\}$ and an independent subset of the $c_a$ randomly over a finite field. The corresponding anti-holomorphic shift can be constructed by taking the parity conjugate.

Let us now consider how a holomorphic shift affects the set of denominator factors $\mathcal{D}$ in eq. (58). Firstly, the holomorphic spinor products are linear functions of $t$

$$\langle ab \rangle \to \langle ab \rangle + t \left( c_a \langle \eta b \rangle + c_b \langle a \eta \rangle \right), \quad (62)$$

whereas the anti-holomorphic spinor products remain unchanged. It is easy to see that the curve parametrised by $t$ intersects each of the ten distinct codimension-one varieties $V(\langle ab \rangle)$ as well as each of the fifteen varieties $V(\langle a|b+c|a])$. Nevertheless, it does not intersect any of the ten varieties $V([ab])$. By comparing the form of the $\mathcal{D}_j$ on this univariate slice with the

denominator of $r_i$ on the slice, one can compute the exponents $q_{ij}$ corresponding to the $\mathcal{D}_j$ which are not purely anti-holomorphic (i.e. the $[ab]$). In order to determine the exponents of the $[ab]$ denominator factors, we repeat the procedure on the anti-holomorphic shift. We note that the anti-holomorphic shift must yield the same $q_{ij}$ for $\langle a|b+c|a]$ as the holomorphic one, providing a consistency check. Having determined the $q_{ij}$ exponents in eq. (54), we can now determine the mass dimension and spinor weight of the numerators $\mathcal{N}_i$, allowing us to write an ansatz for them in terms of spinor-helicity variables.

**Improved Ansatz**    It is well known that the $r_i$ are not all independent rational functions, and it is sufficient to reconstruct a basis of this space of functions. We begin by sorting the $r_i$ by the mass dimension of their numerator $\mathcal{N}_i$, and then apply standard linear-algebra techniques on finite-field-valued evaluations of the $r_i$ to determine a basis of the function space of the $r_i$. We therefore express the pentagon-function coefficients as

$$r_i = \tilde{r}_j M_{ji}, \tag{63}$$

where $M_{ji}$ is a rectangular matrix with $M_{ji} \in \mathbb{Q}$. This observation allows us to reduce the number of functions to reconstruct.

A further important simplification comes from the observation that the rational coefficients in a scattering amplitude are less naturally expressed in common-denominator form, and should in fact be cast in some partial-fractions representation of the form

$$\tilde{r}_i = \sum_k \tilde{r}_{ik}, \qquad \text{with} \qquad \tilde{r}_{ik} = \sum_k \frac{\mathcal{N}_{ik}}{\prod_j \mathcal{D}_j^{q_{ijk}}}, \tag{64}$$

where the $\mathcal{N}_{ik}$ are polynomials in spinor brackets and the $q_{ijk}$ are integer exponents that determine the exact partial-fractions decomposition. For analytic reconstruction approaches, this means that in general a common-denominator ansatz is unnecessarily large. However, systematically constructing a compact partial-fractions ansatz for a generic amplitude remains a challenging task (see refs. [35, 36, 82] for recent progress in this area). A practical solution to this issue is to explore several possible partial-fractions decompositions, determine the size of the ansatz of the $\mathcal{N}_{ik}$ for each of them, and declare that a suitable (if not optimal) decomposition has been found when the ansatz is small enough for the $\mathcal{N}_{ik}$ to be reconstructed. For the amplitudes we are concerned with in this paper, however, we were able to identify a very convenient partial-fractions decomposition that we now discuss.

To construct compact ansätze for the rational functions, we build upon the observation that for several one-loop five-point massless amplitudes all poles of the form $\langle c|a+b|c]^\alpha$ can be separated into different fractions (see e.g. [83]). This feature is related to the fact that the poles $\langle c|a+b|c]$ are spurious, and can be used to greatly simplify analytic expressions for amplitudes [84]. Given this observation, we work under the assumption that a one-loop-like partial-fractions representation exists at two loops. The validity of this assumption will be tested during the calculation. To take an explicit example, consider a function whose common-denominator form reads

$$\frac{\mathcal{N}}{\langle 1|2+4|1]\langle 1|2+5|1]\langle 2|1+5|2]\langle 4|1+2|4]^2\langle 4|1+5|4]^2\langle 5|1+2|5]^2\langle 5|1+4|5]^2},$$

where $\mathcal{N}$ is an unknown polynomial in spinor brackets to be determined. The partial-fractions decomposition would then take the form

$$\begin{aligned}
&\frac{\mathcal{N}_1}{\langle 1|2+4|1]} + \frac{\mathcal{N}_2}{\langle 1|2+5|1]} + \frac{\mathcal{N}_3}{\langle 2|1+5|2]} + \frac{\mathcal{N}_4}{\langle 4|1+2|4]^2} \\
&+ \frac{\mathcal{N}_5}{\langle 4|1+5|4]^2} + \frac{\mathcal{N}_6}{\langle 5|1+2|5]^2} + \frac{\mathcal{N}_7}{\langle 5|1+4|5]^2},
\end{aligned} \tag{65}$$

where the $\mathcal{N}_i$ are unknown polynomials in spinor brackets. The ansätze for each of them is substantially simpler than that for the numerator $\mathcal{N}$ in the common-denominator form, making the reconstruction substantially more efficient.

Another layer of simplification of the reconstruction procedure comes from the observation that, when considering a partial-fractions representation, many of the basis functions are *partially* contained within the vector space spanned by the others. More precisely, given a basis element $\tilde{r}_n$ expressed in the form of eq. (64), many of the $\tilde{r}_{nk}$ belong to span($\tilde{r}_{i\neq n}$). If we consider the reconstruction of a basis function $\tilde{r}_n$, this motivates us to include other basis functions in its ansatz, while dropping many of the terms in the partial-fractions decomposition. That is, if we include other basis elements $\tilde{r}_{i\neq n}$ when constructing an ansatz for $\tilde{r}_n$, then it will suffice to supplement this set of functions with a set that parametrises only the part of $\tilde{r}_n$ not contained in span($\tilde{r}_{i\neq n}$). Naturally, there is a family of such ansätze, with the members corresponding to different choices of terms in the partial-fractions decomposition to include. Concretely, we consider a family of ansätze, where each member is of the form

$$\tilde{r}_n = \frac{\tilde{\mathcal{N}}_n}{\prod_j \mathcal{D}_j^{\tilde{q}_{nj}}} + \sum_{i\neq n} c_{ni}\tilde{r}_i, \tag{66}$$

where $\tilde{\mathcal{N}}_n$ is an unknown polynomial in spinor brackets and the $c_{ni}$ are unknown rational numbers. The exponents $\tilde{q}_{nj}$ in eq. (66) are chosen so that the involved denominator factors are a subset of those in the common denominator form of $\tilde{r}_n$. Given our one-loop-like motivation, we choose the $\tilde{q}_{nj}$ so that only one pole of the form $\langle a|b+c|a]^\alpha$ is involved in each member of the family. In the example of eq. (65), there are seven members of the family of ansätze, each corresponding to a term in eq. (65). Taking each member in turn, we sample the $\tilde{r}_n$ appropriately using finite-field kinematics randomly generated with lips [85, 86]. We then attempt to fit $\tilde{r}_n$ with each ansatz, until we find that one is successful, validating our working assumption. If any of the working assumptions we made were not valid we would not find a successful fit, but in all cases considered in this work we find that our assumption holds. We note that an important step in this procedure is to parametrise polynomials in spinor brackets. This is a non-trivial exercise due to momentum conservation and Schouten identities and we make use of the systematic solution proposed in ref. [36].

Having constrained a single $\tilde{r}_n$ with an ansatz of the form given in eq. (66), we have determined the $\tilde{\mathcal{N}}_n$, the $\tilde{q}_{nj}$ and the $c_{ni}$. As we do not yet know the analytic form of the $\tilde{r}_{i\neq n}$, we have not yet determined the analytic form of $\tilde{r}_n$. Nevertheless, we can sidestep this problem by choosing to change basis and replace $\tilde{r}_n$ with the first term in eq. (66). In order to fully determine a complete set of linearly independent rational functions, we then apply this procedure iteratively. Once this procedure is finished we will need a different matrix $M_{ji}$ that relates our basis to the original set of rational functions $r_i$ in eq. (63). This matrix can be obtained in a similar way as $M_{ji}$ was determined.

Let us finish by briefly commenting on the benefits of our ansatz procedure. We find that the dimension of each ansatz that we make is greatly reduced in comparison to common denominator form. Indeed, the improved ansatz in eq. (66) involves a numerator of much lower mass dimension than that in common denominator form, supplemented only by a few extra parameters for the other rational functions. In practice we find that the largest ansatz we use has only $\mathcal{O}(4000)$ unknowns, which is approximately 10 times smaller than the largest ansatz in common-denominator form. Importantly, as we systematically walk through all possible choices of ansätze we can recycle the numerical evaluations. Furthermore, given the small number of unknowns in the improved ansatz, the many linear systems that we solve are performed at negligible computational cost in comparison to the numerical sampling. Finally, we note that once a successful small ansatz for an $\tilde{r}_n$ of the form in eq. (66) has been found,

we can efficiently further simplify the analytic form of the $\tilde{r}_n$ by systematically trying simpler ansätze with fewer poles and potentially partial fractioning.

**Mandelstam reconstruction**   We also perform the reconstruction computation in Mandelstam invariants following the approach of ref. [47]. The pentagon-function coefficients are split into parity-even and parity-odd parts, and are normalized by the corresponding tree, or a spinor-weight factor in case the latter vanishes. The denominators in Mandelstam invariants are obtained with the same procedure described above, with the difference that a single uni-variate slice now suffices since the denominators are little-group invariant (given by products of alphabet letters). We sort the coefficients by ansatz size, and select the simplest subset which constitutes a basis of the vector space of the rational functions. The numerator ansatz for the most complicated parity-even or parity-odd coefficient is now a polynomial of degree 32 in 5 independent variables, corresponding to 58905 unknown parameters. The number of evaluations of the remainders required to fit the free parameters of this ansatz is twice this number, since for each kinematic point we also require an evaluation at the parity conjugate point in order to differentiate the parity-even from the party-odd parts of the pentagon-function coefficients. In conclusion, to fit the ansatz in common-denominator form the number of required evaluations is 117810 in Mandelstam invariants, compared to 29059 in spinor variables. We verified numerically over both $\mathbb{C}$ and $\mathbb{F}_p$ that the results obtained with the two computations match.

## 4   Results

### 4.1   Efficiency Of Analytic Reconstruction

In order to show the impact of our reconstruction procedure on the efficiency of the calculation, we gather here representative data at various stages of the calculation. In table 1, we display a summary of the complexity of the amplitudes in common denominator form when making use of spinor-helicity variables. Extending the notation of eq. (56), we denote the collective little-group weights of $\mathcal{N}_i$ as $\{\mathcal{N}_i\}$. For comparison, in the same table, we also reproduce the corresponding information for the planar finite remainders previously obtained in ref. [48]. In table 2, we summarize the size of the improved ansatz. For each remainder, we provide the mass-dimension and little-group weights of the most complicated rational function. We use this information to construct the ansätze with the algorithm presented in ref. [36]. The Gröbner-basis computation is performed with SINGULAR [74]. Enumeration of the spinor monomials is performed with OR-TOOLS CP-SAT [87]. The resulting dimension of the ansatz can therefore easily be counted, and we provide this information in the final column of our tables. We note that this cannot be determined from the mass dimension alone.

In summary, we see that reconstruction of the non-planar finite remainders with the original common-denominator ansatz requires about four times more samples than in the planar case. Furthermore, we see that the sampling requirement for the non-planar computation is strongly reduced when we use the improved ansatz of eq. (66). In fact, with $\mathcal{O}(4000)$ required samples, the complexity is below that of the common-denominator ansatz of the planar amplitudes. We note that for the all-plus configurations, the ansatz size is unchanged with respect to table 1 as the most complex function does not contain a $\langle c|a+b|c]$ pole and so is unaffected by the procedure of section 3.3. After the analytic reconstruction, we perform further clean-up following the partial-fraction strategies of refs. [35, 36] and simplify the rational functions to the point where none has more than about 100 free coefficients. The file size of the final results is then dominated by the matrices of rational numbers, while the rational functions are about

| | Contribution | $\dim(\mathrm{span}(r_i))$ | $\max_j([\mathrm{Num}(\tilde{r}_j)]), \{\mathrm{Num}(\tilde{r}_{j_{max}})\}$ | Common Den. Ansatz Size |
|---|---|---|---|---|
| non-planar | $R^{(2,1)}_{-++}$ | 174 | 48, {1, -3, -6, 2, 2} | 29059 |
| | $R^{(2,\tilde{N}_f)}_{-++}$ | 88 | 47, {4, 4, -5, 3, 4} | 24582 |
| | $R^{(2,1)}_{+++}$ | 49 | 21, {5, 4, 3, 3, 3} | 1092 |
| | $R^{(2,\tilde{N}_f)}_{+++}$ | 24 | 20, {2, 4, 6, 6, 6} | 535 |
| planar | $R^{(2,0)}_{-++}$ | 87 | 35, {-3, 0, 6, -3, -2} | 7358 |
| | $R^{(2,N_f)}_{-++}$ | 29 | 15, {-2, -2, 0, -3, -3} | 378 |
| | $R^{(2,0)}_{+++}$ | 31 | 20, {-2, -4, -2, -2, -2} | 1140 |
| | $R^{(2,N_f)}_{+++}$ | 6 | 8, {1, 3, 1, 1, 2} | 44 |

Table 1: Summary of rational function space in common denominator form for the non-planar finite remainders (this work), and, for reference, for the planar ones (originally calculated in ref. [48]). In the first column, we label the associated finite remainder. In the second, we give the number of linearly independent rational functions that arise in the remainder. In the third, we record the mass dimension and little group weights of the most complicated rational function in the basis. In the final column, we state the number of terms in the common-denominator ansatz for this function.

one order of magnitude smaller. We discuss the ancillary files in more detail in section 4.3.

Let us now briefly comment on the analytic properties of the amplitudes. Firstly, we verified that all iterated integrals with the letter $W_{31} = \mathrm{tr}_5$ cancel out in all finite remainders. Secondly, we find that only 162 independent combinations of irreducible weight-four functions are required to express $\mathcal{H}^{(2)}$, a number that is significantly lower than the 472 required to span the full space of irreducible weight-four pentagon functions [56]. It is also interesting to compare the functions arising in the planar triphoton amplitudes with those for leading-color three-jet production. We find that only 48 independent combinations of irreducible weight-four functions are required to express the planar triphoton amplitudes and that they span a proper subspace of those needed for three-jet production.

## 4.2 Validation

In order to validate our computation, we have performed various checks at intermediate stages, as well as on the final result which we now discuss. To verify the surface terms, we produced numerical reduction tables on one phase-space point using FIRE6 [88] and checked that the surface terms reduce to zero. The evaluation of the amplitudes with the numerical unitarity method was carried out by the well-tested code CARAVEL [31]. The construction of finite remainders was also carried out within CARAVEL, thus confirming the expected pole-structure on every phase-space point. In particular we confirm that the contribution $A^{(2,\tilde{N}_f)}$ is finite, which is expected as its coupling structure is non-zero for the first time at two loops. This furthermore implies that this contribution is invariant under scale variations, which we also confirm. To validate the reconstruction methodology, we confirm that the analytic results reconstructed in spinor variables agree with the results obtained using established Mandelstam-variable reconstruction techniques. Furthermore, we check that the analytic results match numerical evaluations from CARAVEL on finite fields of different characteristic. For the planar contributions $R^{(2,0)}$ and $R^{(2,N_f)}$, we find full agreement between our result and a previous calculation [48]. Finally, we evaluated the remainders in a number of collinear configurations

| Contribution | max($[\tilde{\mathcal{N}}]$), $\{\tilde{\mathcal{N}}\}$ | Improved ansatz Size |
|:---:|:---:|:---:|
| $R^{(2,1)}_{-++}$ | 30, {1, -3, -6, 2, 2} | 4003 |
| $R^{(2,\tilde{N}_f)}_{-++}$ | 31, {4, 4, -5, 3, 4} | 3810 |
| $R^{(2,1)}_{+++}$ | 21, {5, 4, 3, 3, 3} | 1092 |
| $R^{(2,\tilde{N}_f)}_{+++}$ | 20, {2, 4, 6, 6, 6} | 535 |

Table 2: Summary of size of the function space of the numerator $\tilde{\mathcal{N}}$ in the improved ansatz of eq. (66) that was used to reconstruct the analytic results. In the second column, we record the mass dimension and little-group weights of the most complicated numerator function in the basis. In the last column, we state the number of terms in the numerator ansatz for this function. By comparison to table 1, we see that the improved ansatz has greatly decreased the number of required samples.

to verify the expected singular behavior. In particular, we validate the sign of the $R^{(2,\tilde{N}_f)}$ contribution by numerically checking that its behavior in collinear limits is consistent with the universal expectation. To this end, we make use of the known two-loop four-point amplitudes of ref. [52].

## 4.3 Ancillary Files

In the ancillary files associated to this paper, we present all finite remainders through two loops. All finite remainders are presented in the form

$$R = \tilde{r}_j M_{ji} h_i\,, \tag{67}$$

where $\tilde{r}_j$ are rational functions, $M_{ji}$ is a matrix of rational numbers and the $h_i$ are monomials of pentagon functions. The ancillary files are provided in `Mathematica` format, organized by gauge-invariant contributions, with naming conventions based on couplings and helicities.

To aid with the understanding of the ancillary files, we provide assembly scripts that evaluate the results of this paper and reproduce the benchmark values in appendix B, which are also found in the file `targets.m`. For ease of use, we provide these assembly scripts both in `Mathematica` (`amp_eval.m`) and in Python (`amp_eval.py`) format. The Python tests can be run with `pytest` [89]. The assembly scripts make use of a series of raw analytic files for each amplitude. Each finite remainder is associated to a subfolder of `anc/amplitudes` which contains the following files:

- `BasisCoefficients.m`: The rational functions $\tilde{r}_j$ in eq. (67). Each element of the list is either a rational function of spinor variables or a list of integers. A list of integers represents a permutation of the spinors in the previous element in the list. For a list of integers {i1, i2, i3, i4, i5} the permutation to be applied is

$$(i_1 i_2 i_3 i_4 i_5 \rightarrow 12345).$$

- `Matrix.m`: The matrix of rational numbers $M_{ji}$ in eq. (67). The matrices are given in the sparse coordinate list (COO) format. In this format, only the non-zero values are stated and all unspecified entries are implicitly zero. The non-zero entries in COO format are specified as {row index, column index} -> value.

- `BasisPentagons.m`: The pentagon functions in eq. (67).

Cached values for the pentagon-function monomials at the kinematic point in appendix B (see eq. (83)) are given in `FValues.m`. The files `Benchmark.m` contain target values for the individual pentagon-function coefficients $r_i = \tilde{r}_j M_{ji}$ at this kinematic point.

**Explicit example:** $R_{+++}^{(2,\tilde{N}_f)}$ In order to elucidate the structure of the results accompanying this paper, we discuss in detail the structure of one of the non-planar finite remainders, $R_{+++}^{(2,\tilde{N}_f)}$. The dimension of the rational function space is 24 (see table 1). The number of contributing pentagon-function monomials is 71. The sparse rational matrix $M$ has dimension $24 \times 71$, with 177 nonzero entries. The basis of rational functions $\tilde{r}$ can be written as

$$
\begin{aligned}
\tilde{r} = \Big\{ &\tilde{r}_1 \,,\, \tilde{r}_1\big|_{453 \to 345} \,,\, \tilde{r}_1\big|_{534 \to 345} \,,\, \tilde{r}_4 \,,\, \tilde{r}_4\big|_{453 \to 345} \,,\, \tilde{r}_4\big|_{534 \to 345} \,,\, \tilde{r}_4\big|_{543 \to 345} \,,\, \tilde{r}_4\big|_{435 \to 345} \,,\\
&\tilde{r}_4\big|_{354 \to 345} \,,\, \tilde{r}_{10} \,,\, \tilde{r}_{10}\big|_{453 \to 345} \,,\, \tilde{r}_{10}\big|_{534 \to 345} \,,\, \tilde{r}_{13} \,,\, \tilde{r}_{13}\big|_{453 \to 345} \,,\, \tilde{r}_{13}\big|_{534 \to 345} \,,\, \tilde{r}_{16} \,,\\
&\tilde{r}_{16}\big|_{453 \to 345} \,,\, \tilde{r}_{18} \,,\, \tilde{r}_{18}\big|_{453 \to 345} \,,\, \tilde{r}_{18}\big|_{534 \to 345} \,,\, \tilde{r}_{18}\big|_{543 \to 345} \,,\, \tilde{r}_{18}\big|_{435 \to 345} \,,\, \tilde{r}_{23} \,,\, \tilde{r}_{24} \Big\} \,,
\end{aligned}
\tag{68}
$$

where we make explicit that many of the basis functions are obtained by permuting momenta in previous elements. It is interesting that owing to the symmetries of the rational function space it suffices to give 8 explicit spinor-helicity expressions. The basis elements that generate the others under permutations are given by

$$
\begin{aligned}
\tilde{r}_1 &= 16 \frac{[13]^2}{[12]\langle 45 \rangle^2} \,,\\[4pt]
\tilde{r}_4 &= -8 \frac{\langle 12 \rangle [13][34](\langle 34 \rangle \langle 12 \rangle + 2\langle 24 \rangle \langle 13 \rangle)}{\langle 14 \rangle^2 [14] \langle 15 \rangle \langle 25 \rangle \langle 34 \rangle} \,,\\[4pt]
\tilde{r}_{10}^D &= -24 \frac{\langle 12 \rangle \langle 24 \rangle [34]}{\langle 14 \rangle \langle 15 \rangle \langle 25 \rangle \langle 34 \rangle} \,, \quad \tilde{r}_{10}^S = -16 \frac{\langle 12 \rangle [15][34]}{\langle 15 \rangle \langle 34 \rangle \langle 1|2+5|1]} \,,\\[4pt]
\tilde{r}_{10} &= \tilde{r}_{10}^D + \tilde{r}_{10}^D\big|_{345 \to 435} + \tilde{r}_{10}^S \,,\\[4pt]
\tilde{r}_{13}^D &= 24 \frac{\langle 12 \rangle [15]}{\langle 13 \rangle \langle 34 \rangle \langle 45 \rangle} - 48 \frac{[15]\langle 25 \rangle^2}{\langle 24 \rangle \langle 35 \rangle^2 \langle 45 \rangle} + 24 \frac{\langle 12 \rangle^2 [15]}{\langle 13 \rangle \langle 14 \rangle \langle 23 \rangle \langle 45 \rangle} \,,\\[4pt]
\tilde{r}_{13}^S &= 48 \frac{[15]\langle 24 \rangle}{\langle 34 \rangle^2 \langle 45 \rangle} - 24 \frac{[15]\langle 25 \rangle}{\langle 34 \rangle \langle 35 \rangle \langle 45 \rangle} \,,\\[4pt]
\tilde{r}_{13} &= \tilde{r}_{13}^D + \tilde{r}_{13}^D\big|_{345 \to 435} + \tilde{r}_{13}^S \,,\\[4pt]
\tilde{r}_{16}^D &= -48 \frac{\langle 23 \rangle^2 [23]}{\langle 15 \rangle \langle 34 \rangle^2 \langle 35 \rangle} + 24 \frac{[23]\langle 24 \rangle \langle 25 \rangle}{\langle 15 \rangle \langle 34 \rangle \langle 45 \rangle^2} \,,\\[4pt]
\tilde{r}_{16}^S &= 16 \frac{\langle 12 \rangle [13][45]}{\langle 13 \rangle \langle 45 \rangle \langle 1|2+3|1]} - 48 \frac{\langle 12 \rangle \langle 23 \rangle [23]}{\langle 14 \rangle \langle 15 \rangle \langle 34 \rangle \langle 35 \rangle} \,,\\[4pt]
\tilde{r}_{16} &= \tilde{r}_{16}^D + \tilde{r}_{16}^D\big|_{345 \to 354} + \tilde{r}_{16}^S \,,
\end{aligned}
\tag{69}
$$

where the superscripts refer to doublets (D) and singlets (S) under the permutation transformation used to express the coefficient. We omit the three functions $\tilde{r}_{18}, \tilde{r}_{23}$ and $\tilde{r}_{24}$ as they are too complicated to print in the text.

## 4.4 Numerical Evaluation

To facilitate the use of our results in phenomenological applications, we have implemented our analytic expressions into the C++ library `FivePointAmplitudes` [49], making use of `PentagonFunctions++` [56] for the numerical evaluation of the pentagon functions. For use in such applications, it is important that our implementation is capable of producing numerically-stable results while maintaining reasonable evaluation times. To demonstrate the numerical performance of our implementation, we study the (helicity) finite remainders and

the hard function $\mathcal{H}$ on a sample of 100$k$ phase-space points generated by MATRIX v2 [90]. We adopt the phase-space definition from ref. [13], and we set the renormalization scale to $\mu = m_{\gamma\gamma\gamma}$.

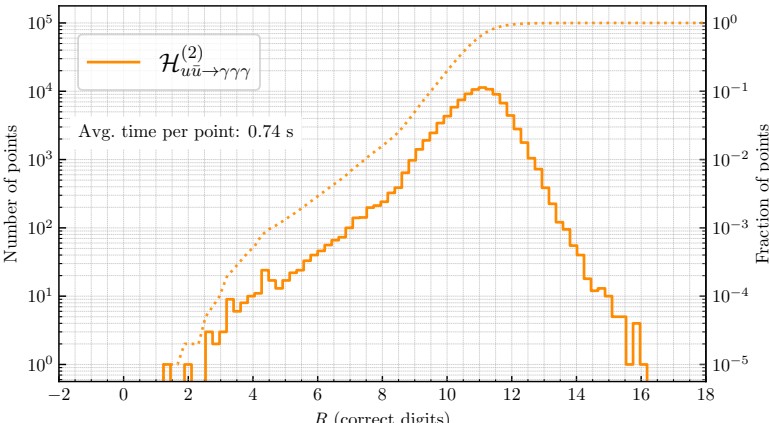

(a) Distributions of correct digits for full $\mathcal{H}^{(2)}$ with five active flavours, i.e. $N_c = 3$, $N_f = 5$. The dashed line shows the cumulative distribution.

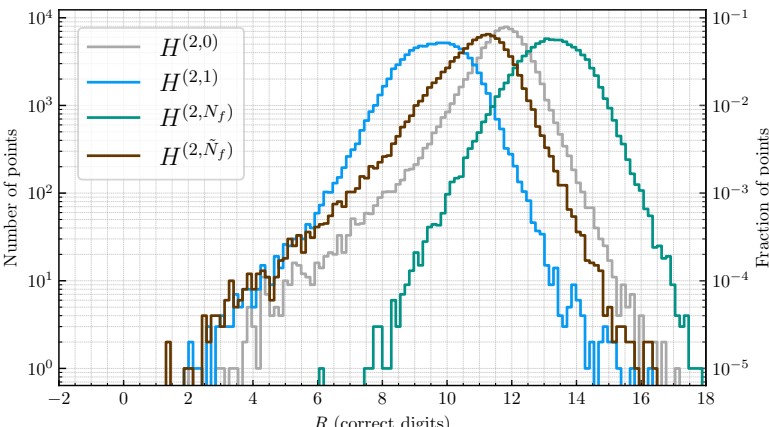

(b) Relative error of the contributions defined in eq. (20) to the hard function $\mathcal{H}^{(2)}$.

Figure 3: Distributions of correct digits $R$ (see eq. (70)) that characterize the numerical performance of the C++ implementation of our analytic results in FivePointAmplitudes [49]. The phase-space sample of 100$k$ phase-space points is defined in ref. [13] and generated by MATRIX v2 [90]. The renormalization scale is set to $\mu = m_{\gamma\gamma\gamma}$.

We characterize the numerical stability of our implementation in fig. 3, where we plot the distributions of correct digits $R$,

$$R := -\log_{10}\left|1 - \frac{X_{\text{double}}}{X_{\text{quad}}}\right|, \tag{70}$$

for various quantities $X$. In this way, we use a quadruple-precision evaluation ($X_{\text{quad}}$) to calculate the accuracy of the double-precision evaluation ($X_{\text{double}}$) on each point. To catch and correct unstable evaluations we recycle the precision rescue system developed in ref. [91] for

the amplitudes describing three-jet production at hadron colliders. We note however that in this case it triggers only on a few points from the whole sample, so its effect on the evaluation time is insignificant. In fig. 3a we show the $R$-distribution for the hard function $\mathcal{H}$ defined in eq. (20) for the dominant partonic process $u\bar{u} \to \gamma\gamma\gamma$ with five active massless flavors, i.e. $N_c = 3$, $N_f = 5$. We observe overall excellent numerical stability, and the average evaluation time of less than a second on a single CPU core. We can therefore conclude that our implementation is suitable for phenomenological applications. Since the analytic complexity of the subleading-color contributions is notably higher than that of the leading contributions, it is of interest to compare their relative numerical stability. In fig. 3b we show the $R$-distributions for each of the four contributions to $\mathcal{H}$ separately. Indeed, we observe that the subleading functions $H^{(2,1)}$ and $H^{(2,\tilde{N}_f)}$ are slightly less numerically stable than the leading functions $H^{(2,0)}$, $H^{(2,N_f)}$. Nevertheless, their numerical behavior is clearly adequate for the anticipated phenomenological applications.

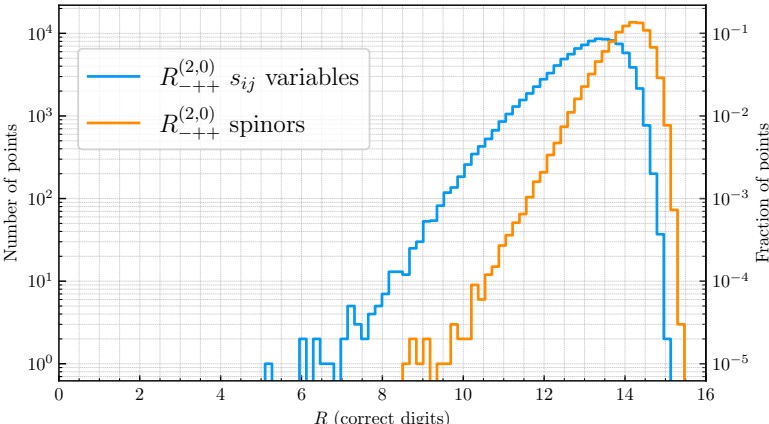

Figure 4: Numerical stability of the rational functions contributing to the finite remainder $R^{(2,0)}_{-++}$ in two different representations: one in terms of Mandelstam invariants and the other in terms of spinor-helicity variables. At each phase-space point, we plot the lowest number of correct digits $R$ over the pentagon-function coefficients $r_i$ of eq. (22). See the caption of fig. 3 for details of the phase-space definition.

Finally, let us recall that we employ a spinor representation of rational coefficients in our numerical implementation. Nevertheless, as discussed in section 3.3, we also reconstructed the rational coefficients in terms of Mandelstam variables. With this data at hand, it is interesting to study the impact of the choice of variables on the numerical stability of the analytic expressions. In fig. 4 we show the $R$-distributions for the coefficients $r_i$ of pentagon functions as defined in eq. (22). Specifically, on each phase-space point we compute the value of $R$ for each $r_i$ and plot $\min_{r_i}(R)$. As a representative example, we show this distribution for the finite remainder $R^{(2,0)}_{-++}$, first using the compact spinor representation of the ancillary files, and then in a representation in terms of Mandelstam variables that has been put into a partial fractions representation using MultivariateApart [92]. We see that the tail of the distribution of $R$ in the spinor case is improved by about two digits with respect to the Mandelstam case, a behavior that we find is consistent across all helicity amplitudes. It is therefore evident that the spinor representation improves the numerical stability compared to the representation through Mandelstam invariants. In spite of this, for the contribution $H^{(2,0)}$ we observe overall similar behavior as in ref. [48]. This implies that the numerical accuracy of $\mathcal{H}^{(2)}$ is limited by the accuracy of the pentagon-function evaluation.

## 5 Subleading Color Corrections To Hard Function

In all phenomenological studies of triphoton production at hadron colliders to date, the double-virtual NNLO QCD corrections have been included in the leading-color approximation. Given our results for the subleading color contributions to the double-virtual corrections, it is interesting to consider the numerical impact of these corrections. Specifically, in refs. [12, 13] the hard function was taken to be

$$
\mathcal{H}^{(2)} \to \mathcal{H}^{(2)}_{\text{l.c.}} := \frac{N_c^2}{4} H^{(2,0)},
\tag{71}
$$

instead of the full result in eq. (20). In this section, we study the impact of the subleading-color contributions that we have calculated in this work relative to $\mathcal{H}^{(2)}_{\text{l.c.}}$. We focus on the dominant partonic channel $u\bar{u} \to \gamma\gamma\gamma$, and we consider distributions of the relative sizes of corrections with respect to the leading-color result,

$$
\delta \mathcal{H}^{(2)}_x = \frac{\Delta \mathcal{H}^{(2)}_x}{\mathcal{H}^{(2)}_{\text{l.c.}}},
\tag{72}
$$

over the same phase-space points used in section 4.4. Here $\Delta \mathcal{H}^{(2)}_x$ is one of the three corrections in eq. (20):

$$
\begin{aligned}
\Delta \mathcal{H}^{(2)}_{N_c} &:= -\frac{1}{4}(H^{(2,0)} + H^{(2,1)}) + \frac{1}{4N_c^2} H^{(2,1)}, \\
\Delta \mathcal{H}^{(2)}_{N_f} &:= C_F T_F N_f H^{(2,N_f)}, \\
\Delta \mathcal{H}^{(2)}_{Q_f^2} &:= C_F T_F \left( \sum_{f=1}^{N_f} Q_f^2 \right) H^{(2,\tilde{N}_f)},
\end{aligned}
\tag{73}
$$

such that $\mathcal{H}^{(2)} = \mathcal{H}^{(2)}_{\text{l.c.}} + \Delta \mathcal{H}^{(2)}_{N_c} + \Delta \mathcal{H}^{(2)}_{N_f} + \Delta \mathcal{H}^{(2)}_{Q_f^2}$, and $\delta \mathcal{H}^{(2)} := \mathcal{H}^{(2)}/\mathcal{H}^{(2)}_{\text{l.c.}} - 1$. We note that the $\delta \mathcal{H}_x$ are scheme dependent, and can vary substantially between different schemes. In fig. 5, we plot the distribution of the $\delta \mathcal{H}_x$ over phase space in the $q_T$, $\overline{\text{MS}}$ and Catani schemes (see appendix A.2 for our scheme definitions). The average correction size in fig. 5 should provide a reasonable proxy for the subleading-color effects in fiducial cross sections, whereas the shape demonstrates how the corrections vary over phase space. Our phase-space sample and conventions are the same as that used in the studies of numerical stability. Specifically, our 100k sample points are taken from the phase space used in the Monte-Carlo cross-section computation of ref. [13], as generated by MATRIX v2 [90]. We work with five active flavors, i.e. $N_f = 5$, and set $N_c = 3$ and the renormalization scale to $\mu = m_{\gamma\gamma\gamma}$.

Let us now discuss key features of these distributions. First, we consider the impact of the choice of IR subtraction scheme and see in fig. 5 that the $q_T$ and $\overline{\text{MS}}$ schemes behave similarly, and both differ noticeably from the Catani scheme. Next, we note that a highly peaked distribution implies a simple correction by a factor in all differential cross sections, while a smeared distribution implies that the corrections are observable-dependent. Focusing specifically on fig. 5a (the schemes used in refs. [12,13]), we see that all corrections are negative, but behave differently over phase space. The correction from the $N_c$-suppressed terms is sharply peaked at about $-10\%$, in good agreement with the common subleading-color behavior. The $H^{(2,\tilde{N}_f)}$ corrections on the contrary demonstrate significant phase-space dependence. On average, the combined correction is about $-35\%$, and the shape of the distribution suggests that for some observables the correction could reach up to about $-50\%$. Nevertheless, taking into account the smallness of the overall contribution of $\mathcal{H}^{(2)}$ to NNLO cross sections, we expect that the

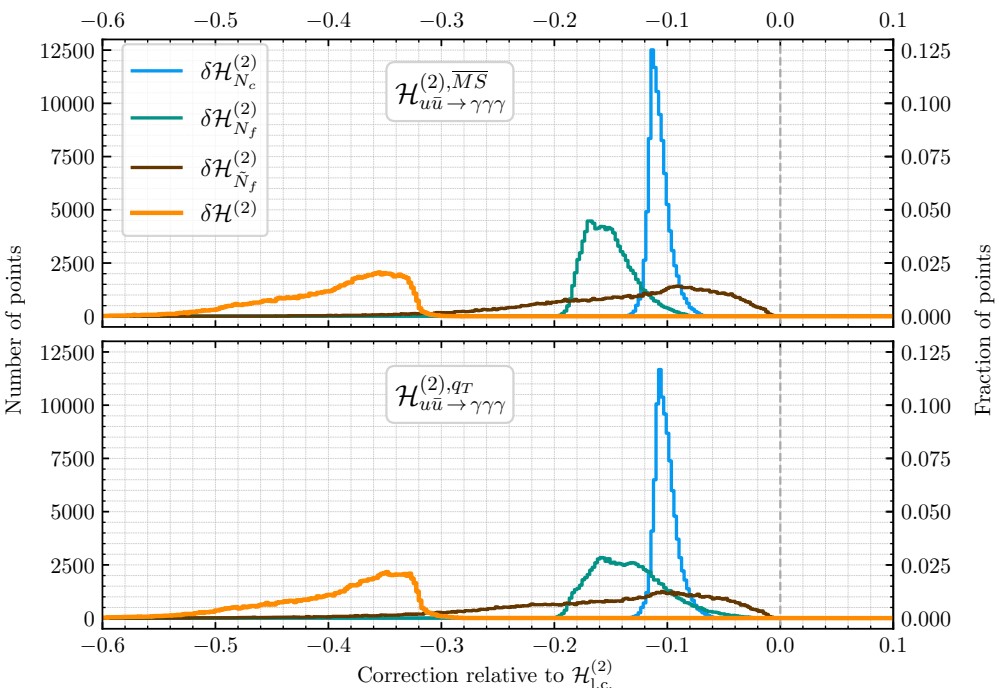

(a) Distribution of subleading-color corrections in the $q_T$ and $\overline{\text{MS}}$ schemes.

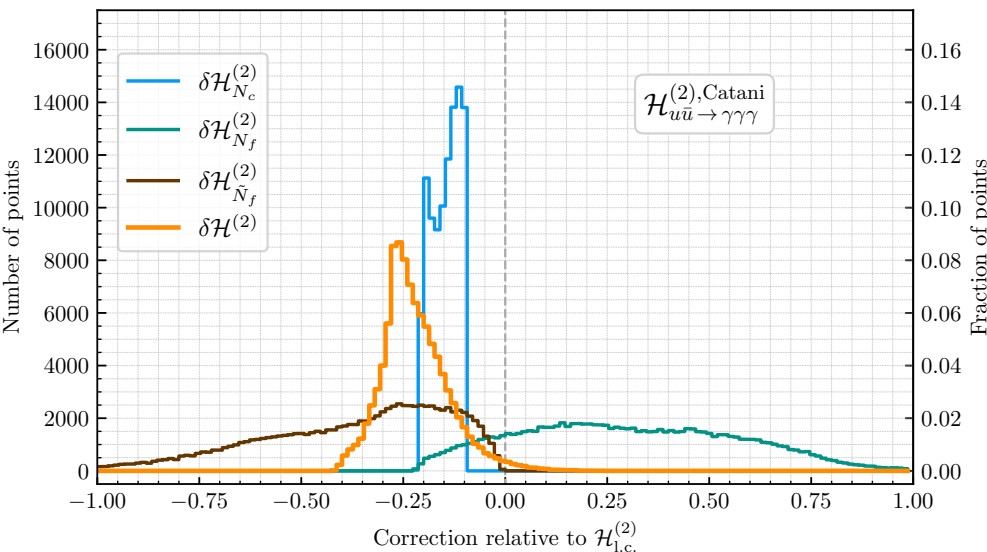

(b) Distribution of subleading-color corrections in the Catani scheme.

Figure 5: Distribution of separate subleading-color corrections $\delta\mathcal{H}_x$ defined in eq. (73) to the the hard function $\mathcal{H}$ alongside the total correction $\delta\mathcal{H}$. The phase-space sample of $100k$ phase-space points is defined in ref. [13] and generated by `MATRIX v2` [90]. The renormalization scale is set to $\mu = m_{\gamma\gamma\gamma}$.

approximation employed in refs. [12, 13] should be largely adequate. Finally, it is worth noting that in the $d\bar{d}$ partonic channel the contribution from $H^{(2,\tilde{N}_f)}$ to $\mathcal{H}^{(2)}$ is four times larger than in $u\bar{u}$, while all other contributions are unchanged. Therefore, $H^{(2,\tilde{N}_f)}$ becomes numerically as important as $\mathcal{H}^{(2)}_{\text{l.c.}}$ in this channel and the approximation of eq. (71) is no longer valid. However, the overall contribution of the $d\bar{d}$ channel to the cross section is highly suppressed by the ratio of quark electric charges, so this effect is invisible.

## 6 Conclusion

In this work, we have computed the complete set of two-loop helicity amplitudes that contribute to the NNLO QCD corrections for triphoton production at hadron colliders, including all previously unknown non-planar contributions. We have derived compact analytic expressions for the two-loop finite remainders that we make available as supplementary material. In addition, we have implemented our analytic results in a C++ library, providing a stable and efficient numerical evaluation that is suitable for phenomenological applications.

To carry out our computation, we employed the numerical unitarity approach as implemented in CARAVEL, which we have suitably extended to take care of the reduction of non-planar five-point integral topologies. To handle the ensuing increase in complexity, we have proposed a new approach for the construction of surface terms that capitalizes on formulating the associated syzygy problem in embedding space, and on the numerical nature of the computational framework. To reconstruct the analytic form of the results from numerical evaluations of the amplitude over finite fields we have constructed an optimized ansatz in spinor-helicity variables. This resulted in a more efficient reconstruction procedure, with the most complicated ansatz having only around 4000 free parameters. We anticipate that the advancements in both of these technical aspects will greatly assist in computations of two-loop amplitudes with more kinematic scales that currently pose significant computational challenges.

Using our results, we have estimated the effect of including the subleading-color corrections in the double-virtual contributions to the cross section. We have found that, on average, over a representative sample of phase space, the two-loop hard function receives an extra negative contribution of around 25-35%. The exact effect is observable and scheme dependent. Due to an overall small contribution from double-virtual corrections to the NNLO cross sections [12, 13], we conclude that the approximation employed in ref. [12–14] should be valid within a few percent, although in practice it is important to verify this conclusion for each observable. Nevertheless, it is clear that in view of the excellent numerical stability and efficiency of our results, including the subleading-color corrections to the double-virtual contributions in future phenomenological studies of triphoton production is straightforward.

## Acknowledgements

VS gratefully acknowledges the computing resources provided by the Max Planck Institute for Physics.

**Funding information** The work of BP was supported by the European Union's Horizon 2020 research and innovation program under the Marie Sklodowska-Curie grant agreement No.896690 (LoopAnsatz). MK's work was funded by the German Research Foundation (DFG) within the Research Training Group GRK 2044. VS has received funding from the European Research Council (ERC) under the European Union's Horizon 2020 research and innovation programme grant agreement 101019620 (ERC Advanced Grant TOPUP).

# A  IR Renormalization

## A.1  Conventions In Catani Scheme

The remainders $\mathcal{R}$ are defined in eq. (18) using the functions $\mathbf{I}^{(1)}$ and $\mathbf{I}^{(2)}$ [51, 52] which we quote here for convenience,

$$
\begin{aligned}
\mathbf{I}^{(1)}(\epsilon) &= C_F \frac{e^{\gamma_E \epsilon}}{\Gamma(1-\epsilon)} \left( \frac{1}{\epsilon^2} + \frac{3}{2\epsilon} \right) \left( -\frac{s_{12}}{\mu^2} - i0 \right)^{-\epsilon}, \\
\mathbf{I}^{(2)}(\epsilon) &= \frac{1}{2} \mathbf{I}^{(1)}(\epsilon) \mathbf{I}^{(1)}(\epsilon) - \frac{\beta_0}{\epsilon} \mathbf{I}^{(1)}(\epsilon) + \frac{e^{-\gamma_E \epsilon} \Gamma(1-2\epsilon)}{\Gamma(1-\epsilon)} \left( \frac{\beta_0}{\epsilon} + K \right) \mathbf{I}^{(1)}(2\epsilon) - \mathbf{H}(\epsilon).
\end{aligned}
\tag{74}
$$

In $\mathbf{I}^{(2)}(\epsilon)$, we have introduced the functions

$$
\begin{aligned}
K &= \left( \frac{67}{18} - \frac{\pi^2}{6} \right) C_A - \frac{10}{9} T_F N_F, \qquad \mathbf{H}(\epsilon) = \frac{e^{\gamma_E \epsilon}}{2\epsilon \Gamma(1-\epsilon)} H_q, \\
H_q &= \left( \frac{\pi^2}{2} - 6\zeta_3 - \frac{3}{8} \right) C_F^2 + \left( \frac{13}{2} \zeta_3 + \frac{245}{216} - \frac{23}{48} \pi^2 \right) C_A C_F + \left( \frac{\pi^2}{12} - \frac{25}{54} \right) T_F C_F N_f.
\end{aligned}
\tag{75}
$$

## A.2  IR-Subtraction Scheme Change

Consider two different IR subtraction schemes, in which finite remainders are

$$
\mathcal{R} = \mathbf{I}\mathcal{A}, \qquad \tilde{\mathcal{R}} = \tilde{\mathbf{I}}\mathcal{A},
\tag{76}
$$

where we suppress the helicity labels. We write the difference between the squared finite remainders as

$$
\Delta := |\tilde{\mathcal{R}}|^2 - |\mathcal{R}|^2 = \left( |\tilde{\mathbf{I}}|^2 - |\mathbf{I}|^2 \right) |\mathcal{A}|^2,
\tag{77}
$$

where the absolute values do not include helicity sums. Overall colour factors have been removed as in eq. (8) and no colour sums are implied.

Upon expansion to the second order in $\alpha_s$ we find

$$
\begin{aligned}
\Delta^{(1)} &= \delta^{(1)} |\mathcal{A}^{(0)}|^2, \qquad \delta^{(1)} := 2\,\mathrm{Re}\left\{ \tilde{\mathbf{I}}^{(1)} - \mathbf{I}^{(1)} \right\}, \\
\Delta^{(2)} &= \bar{\delta}^{(2)} |\mathcal{A}^{(0)}|^2 + \delta^{(1)} 2\,\mathrm{Re}\left\{ \mathcal{A}^{(0)\star} \mathcal{A}^{(1)} \right\}, \quad \bar{\delta}^{(2)} := 2\,\mathrm{Re}\left\{ \tilde{\mathbf{I}}^{(2)} - \mathbf{I}^{(2)} \right\} - |\tilde{\mathbf{I}}^{(1)}|^2 + |\mathbf{I}^{(1)}|^2.
\end{aligned}
\tag{78}
$$

We can rewrite $\Delta^{(2)}$ such that it is defined explicitly through finite quantities as

$$
\Delta^{(2)} = \delta^{(2)} |\mathcal{A}^{(0)}|^2 + \delta^{(1)} 2\,\mathrm{Re}\left\{ \mathcal{A}^{(0)\star} \mathcal{R}^{(1)} \right\}, \qquad \delta^{(2)} := \bar{\delta}^{(2)} + \delta^{(1)} 2\,\mathrm{Re}\left\{ \mathbf{I}^{(1)} \right\}.
\tag{79}
$$

Using the definitions in eq. (19), and taking advantage of the factorization of $|\mathbf{I}|^2$ in the sum, we can therefore write the scheme shift for the hard functions as

$$
\begin{aligned}
\tilde{\mathcal{H}}^{(1)} &= \mathcal{H}^{(1)} + \delta^{(1)}, \\
\tilde{\mathcal{H}}^{(2)} &= \mathcal{H}^{(2)} + \delta^{(1)} \mathcal{H}^{(1)} + \delta^{(2)}.
\end{aligned}
\tag{80}
$$

dependence of the remainder definition in eq. (76) enters in a linear way, the conversion formulas apply equally to helicity-summed hard functions $\mathcal{H}^{(L)}$.

Let us consider converting the finite remainders in the Catani scheme, that is employed in this work, to the ones defined in the $\overline{\text{MS}}$ scheme. Taking the definitions of $\tilde{\mathbf{I}} = \mathbf{I}^{\overline{\text{MS}}}$ from

ref. [53],[4] we obtain

$$\delta_{\overline{\text{MS}}}^{(1)} = C_F \left( \frac{7\pi^2}{6} + 3l_\mu - l_\mu^2 \right),$$

$$\delta_{\overline{\text{MS}}}^{(2)} = C_F^2 \left( \frac{49\pi^4}{72} + \frac{7\pi^2}{2} l_\mu + \left( \frac{9}{2} - \frac{7\pi^2}{6} \right) l_\mu^2 - 3l_\mu^3 + \frac{1}{2} l_\mu^4 \right) \quad (81)$$

$$+ C_F C_A \left( \left( \frac{691\pi^2}{108} - \frac{25\pi^4}{144} - \frac{11\zeta_3}{12} \right) + \left( \frac{67}{6} - \frac{157\pi^2}{72} \right) l_\mu + \left( -\frac{233}{36} + \frac{\pi^2}{6} \right) l_\mu^2 + \frac{11}{18} l_\mu^3 \right)$$

$$+ C_F T_F N_f \left( -\frac{56\pi^2}{27} + \frac{\zeta_3}{3} + \left( -\frac{10}{3} + \frac{11\pi^2}{18} \right) l_\mu + \frac{19l_\mu^2}{9} - \frac{2l_\mu^3}{9} \right),$$

where $l_\mu := \log(s_{12}/\mu^2)$.

Similarly, taking the definitions from eqs. (50-60) from ref. [93] for the $q_T$ scheme we get

$$\delta_{q_T}^{(1)} = C_F \pi^2,$$

$$\delta_{q_T}^{(2)} = C_F^2 \left( \left( -12\zeta_3 - \frac{3}{4} + \pi^2 \right) l_\mu + \frac{\pi^4}{2} \right)$$

$$+ C_F C_A \left( \left( 13\zeta_3 + \frac{245}{108} - \frac{67\pi^2}{24} \right) l_\mu - \frac{187\zeta_3}{36} - \frac{7\pi^4}{48} + \frac{1181\pi^2}{216} + \frac{607}{81} \right) \quad (82)$$

$$+ C_F T_F N_f \left( \left( \frac{5\pi^2}{6} - \frac{25}{27} \right) l_\mu + \frac{17\zeta_3}{9} - \frac{97\pi^2}{54} - \frac{164}{81} \right).$$

## B  Reference Values

To facilitate comparison with our results, we present values for the finite remainders evaluated at a specific kinematic point. The phase-space point we use is:

$$p_1^\mu = \{-0.575, -0.575, 0, 0\},$$
$$p_2^\mu = \{-0.575, 0.575, 0, 0\},$$
$$p_3^\mu = \{0.458858239, 0.405584802, 0.207778343, -0.053665747\}, \quad (83)$$
$$p_4^\mu = \{0.231129408, -0.097079562, 0.009377939, -0.209543351\},$$
$$p_5^\mu = \{0.460012351, -0.308505239, -0.217156282, 0.263209099\}.$$

This phase-space point was chosen to match the values given in ref. [48] for the kinematic invariants.

For the spinors, we follow the conventions used, for example, in ref. [85], i.e. we take

$$\lambda_\alpha = \begin{pmatrix} \sqrt{p^0 + p^3}, \\ \frac{p^1 + ip^2}{\sqrt{p^0 + p^3}} \end{pmatrix}, \quad \text{and} \quad \tilde{\lambda}_{\dot{\alpha}} = \left( \sqrt{p^0 + p^3}, \frac{p^1 - ip^2}{\sqrt{p^0 + p^3}} \right). \quad (84)$$

This implies that with real kinematics we have $\lambda_\alpha^* = \tilde{\lambda}_{\dot{\alpha}}$, when the energy is positive, and $\lambda_\alpha^* = -\tilde{\lambda}_{\dot{\alpha}}$, when the energy is negative. In table 3 we provide the evaluations of the finite remainders at the point of eq. (83). In addition, in tables 4 to 6 we show the evaluations of the tree, one-loop and two-loop bare amplitudes, which can be used to derive the results of

---

[4]Explicitly, $\tilde{\mathbf{I}}^{(1)} = -\mathbf{Z}^{(1)}/2$, $\tilde{\mathbf{I}}^{(2)} = ((\mathbf{Z}^{(1)})^2 - \mathbf{Z}^{(2)})/4$, with the $\mathbf{Z}^{(i)}$ defined in ref. [53].

table 3 following the definitions in eqs. (13) to (15) and (18). The one- and two-loop bare amplitudes are normalized by the following factors

$$\Phi_{+++} = \frac{[31]\langle 12\rangle^3\langle 13\rangle}{\langle 14\rangle^2\langle 15\rangle^2\langle 23\rangle^2} \quad \text{and} \quad \Phi_{-++} = A^{(0)}_{-++} = -\frac{\langle 12\rangle\langle 23\rangle^2}{\langle 14\rangle\langle 15\rangle\langle 24\rangle\langle 25\rangle}. \tag{85}$$

The one-loop and two-loop planar remainders reproduce the values of ref. [48]. In this work the finite remainders were not normalized by any spinor weight, hence a little care is needed when comparing with table 6 of ref. [48]. The last four lines are the new subleading-color, non-planar contributions.

| Finite Remainder | Numerical Evaluation |
|---|---|
| $R^{(1)}_{-++}$ | $31.76842068 - 98.20723767\,i$ |
| $R^{(1)}_{+++}$ | $67.16227913 + 20.80380252\,i$ |
| $R^{(2,0)}_{-++}$ | $726.0944727 - 748.8429540\,i$ |
| $R^{(2,0)}_{+++}$ | $1085.896384 + 310.5673949\,i$ |
| $R^{(2,N_f)}_{-++}$ | $-198.2921242 + 257.4649652\,i$ |
| $R^{(2,N_f)}_{+++}$ | $-233.4239202 - 112.3516487\,i$ |
| $R^{(2,1)}_{-++}$ | $-548.9464331 - 12.71176526\,i$ |
| $R^{(2,1)}_{+++}$ | $-197.0006137 - 822.9175634\,i$ |
| $R^{(2,\tilde{N}_f)}_{-++}$ | $35.98395348 - 188.6772927\,i$ |
| $R^{(2,\tilde{N}_f)}_{+++}$ | $321.0180068 + 179.1161201\,i$ |

Table 3: Finite remainder evaluations at the point of eq. (83) with the spinors defined as in eq. (84). The remainders are evaluated in the Catani scheme as defined in the text.

| | $\epsilon^0$ |
|---|---|
| $A^{(0)}_{-++}$ | $6.73642828 + 10.02454999\,i$ |
| $A^{(0)}_{+++}$ | $0$ |

Table 4: Tree amplitude evaluated at the point of eq. (83) with the spinors defined as in eq. (84).

| | $\epsilon^{-2}$ | $\epsilon^{-1}$ | $\epsilon^0$ | $\epsilon^1$ | $\epsilon^2$ |
|---|---|---|---|---|---|
| $A^{(1)}_{-++}/\Phi_{-++}$ | $-1$ | $-3.17428470$ $-3.14159265\,i$ | $-3.43768120$ $-16.69077767\,i$ | $-4.54236420$ $-48.29215997\,i$ | $-28.34154957$ $-104.73071157\,i$ |
| $A^{(1)}_{+++}/\Phi_{+++}$ | $0$ | $0$ | $-122.48761401$ $-218.20999082\,i$ | $-613.16200128$ $-1772.24966259\,i$ | $-1264.78147357$ $-6727.58375625\,i$ |

Table 5: Bare, normalized one-loop amplitudes evaluated at the point of eq. (83).

| | $\epsilon^{-4}$ | $\epsilon^{-3}$ | $\epsilon^{-2}$ | $\epsilon^{-1}$ | $\epsilon^{0}$ |
|---|---|---|---|---|---|
| $A^{(2,0)}_{-++}\big/\Phi_{-++}$ | 0.5 | $2.25761803$ $+3.14159265\,i$ | $-3.31724534$ $+20.90350062\,i$ | $-55.54942677$ $+44.34772278\,i$ | $-248.76993460$ $-87.79211642\,i$ |
| $A^{(2,0)}_{+++}\big/\Phi_{+++}$ | 0 | 0 | $122.48761401$ $+218.20999082\,i$ | $-132.67559542$ $+2049.61318591\,i$ | $-9927.84571218$ $+3575.60761772\,i$ |
| $A^{(2,N_f)}_{-++}\big/\Phi_{-++}$ | 0 | $0.16666667$ | $1.33587268$ $+1.04719755\,i$ | $4.64626451$ $+12.87251436\,i$ | $10.33373683$ $+83.15472522\,i$ |
| $A^{(2,N_f)}_{+++}\big/\Phi_{+++}$ | 0 | 0 | 0 | $81.65840934$ $+145.47332721\,i$ | $895.94750003$ $+2327.53809534\,i$ |
| $A^{(2,1)}_{-++}\big/\Phi_{-++}$ | 0.5 | $3.17428469$ $+3.14159265\,i$ | $3.54092067$ $+26.66308714\,i$ | $-38.30735107$ $+112.07323410\,i$ | $-265.12342759$ $+331.32292317\,i$ |
| $A^{(2,1)}_{+++}\big/\Phi_{+++}$ | 0 | 0 | $122.48761401$ $+218.20999082\,i$ | $316.44565596$ $+2849.71648558\,i$ | $-6265.18093537$ $+17706.12335722\,i$ |
| $A^{(2,\tilde{N}_f)}_{-++}\big/\Phi_{-++}$ | 0 | 0 | 0 | 0 | $-11.30451464$ $-11.18613860\,i$ |
| $A^{(2,\tilde{N}_f)}_{+++}\big/\Phi_{+++}$ | 0 | 0 | 0 | 0 | $-390.31606513$ $-1248.74833374\,i$ |

Table 6: Bare, normalized two-loop amplitudes evaluated at the point of eq. (83).

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
