# Peer review of "Two-Loop QCD Corrections for Three-Photon Production at Hadron Colliders"

_SciPost Physics_

## Round 1 · Referee Report · Anonymous (Referee 1) · 2023-7-24

Report

The article "Two-Loop QCD Corrections for Three-Photon Production at Hadron Colliders" describes the computation of the complete two-loop amplitudes required for NNLO QCD corrections in three-photon production in hadron-hadron collisions. In the light of precision measurements at the LHC, this computation represents a timely and valuable result. Specifically, it overcomes the leading-colour (or planar) approximation used in existing phenomenological studies for this process. A suitable estimation of the impact of the non-planar corrections on cross-sections is provided and found to be substantial on the level of the squared matrix element.

The authors perform the computation in the framework of numerical unitary and the widely adopted finite-field technique to reconstruct the amplitudes in an analytical form. The article provides two central results on the technical side:
Firstly, as implemented in the CARAVEL software, the numerical unitarity approach is extended to non-planar topologies. One crucial ingredient is the construction of surface terms, and solving the corresponding equations is challenging. The authors tackle these by recasting the equations in embedding space, allowing them to control the equations' polynomial degree and solve them systematically. The authors give a detailed description of the relevant derivations and use a clear and concise notation. This result is a significant development step for the technique.
Secondly, the authors discuss the construction of improved Ansätze for the rational functions appearing in the final result to reduce the number of numerical evaluations needed. They employ a parameterization of the rational expression in terms of spinor-helicity variables, reuse structure appearing in one-loop expressions and use partial fractioning to achieve a highly optimized basis gaining an order of magnitude in efficiency.

In summary, the article is relevant, timely and well-written. Besides providing the results for three-photon production, the article is a stepping stone for future applications of the numerical unitarity method to more complicated processes. Therefore, I recommend the publication of the manuscript in SciPost.

---

## Round 1 · Referee Report · Simon Badger (Referee 2) · 2023-8-9

Report

In the article "Two-Loop QCD Corrections for Three-Photon Production at Hadron Colliders" the authors present full colour double virtual corrections in QCD for the first time. The result is relevant for recent experimental measurements of photonic final states at the LHC but the methodology presented is also of importance for precision amplitude computations with similar kinematics.

The paper is well presented including comprehensive validation and numerical performance tests. I have no hesitation in recommending it for publication.

As a minor comment: The relatively large sub-leading colour double virtual corrections are highlighted throughout the article as being larger than expected, ~30-50%. It is also stated that the differential cross sections are dominated be real radiation contributions. It would be useful if the authors could also provide a quantitative estimate of the correction for the NNLO predictions based on the new results presented here. The conclusions highlight that the currently available leading colour predictions are likely sufficient, which is not clear from the phrasing in the abstract.

---

## Editorial Decision

resubmitted